# CD95/Fas ligand mRNA is toxic to cells

Will Putzbach[1†], Ashley Haluck-Kangas[1†], Quan Q Gao[1], Aishe A Sarshad[2], Elizabeth T Bartom[3], Austin Stults[1], Abdul S Qadir[1], Markus Hafner[2], Marcus E Peter[1,3]*

[1]Department of Medicine, Division Hematology/Oncology, Feinberg School of Medicine, Northwestern University, Chicago, United States; [2]Laboratory of Muscle Stem Cells and Gene Regulation, National Institute of Arthritis and Musculoskeletal and Skin Diseases, National Institutes of Health, Bethesda, United States; [3]Department of Biochemistry and Molecular Genetics, Northwestern University, Chicago, United States

**Abstract** CD95/Fas ligand binds to the death receptor CD95 to induce apoptosis in sensitive cells. We previously reported that CD95L mRNA is enriched in sequences that, when converted to si/shRNAs, kill all cancer cells by targeting critical survival genes (*Putzbach et al., 2017*). We now report expression of full-length CD95L mRNA itself is highly toxic to cells and induces a similar form of cell death. We demonstrate that small (s)RNAs derived from CD95L are loaded into the RNA induced silencing complex (RISC) which is required for the toxicity and processing of CD95L mRNA into sRNAs is independent of both Dicer and Drosha. We provide evidence that in addition to the CD95L transgene a number of endogenous protein coding genes involved in regulating protein translation, particularly under low miRNA conditions, can be processed to sRNAs and loaded into the RISC suggesting a new level of cell fate regulation involving RNAi.
DOI: https://doi.org/10.7554/eLife.38621.001

*For correspondence:
m-peter@northwestern.edu

†These authors contributed equally to this work

Competing interests: The authors declare that no competing interests exist.

## Introduction

Activation of CD95/Fas through interaction with its cognate ligand CD95L or receptor-activating antibodies induces apoptosis in sensitive cells (*Suda et al., 1993*). Virtually all research on CD95 and CD95L has focused on the physical interaction between the two proteins and the subsequent protein-based signaling cascades (*Algeciras-Schimnich et al., 2002*; *Fu et al., 2016*; *Nisihara et al., 2001*; *Schneider et al., 1997*). However, we have recently shown that the mRNA of CD95 and CD95L harbor sequences that when converted into small interfering (si) or short hairpin (sh)RNAs, cause massive and robust toxicity in all tested cancer cells. These CD95/CD95L-derived si/shRNAs target a network of survival genes, resulting in the simultaneous activation of multiple cell death pathways through RNA interference (RNAi) in a process we called DISE (Death Induced by Survival gene Elimination) (*Putzbach et al., 2017*). We determined that for a si/shRNA to elicit this form of toxicity, only positions 2–7 of the guide strand, the 6mer seed sequence, are required (*Putzbach et al., 2017*). More recently, a screen of all 4096 6mer seeds revealed that optimal 6mer seed toxicity requires G-rich seeds targeting C-rich regions in the 3'UTRs of survival genes (*Gao et al., 2018*).

In this report, we show that expression of the CD95L mRNA itself is toxic to cells even without prior conversion to small (s)RNAs. This toxicity is independent of the full-length CD95L protein or expression of the CD95 receptor and resembles DISE. The toxicity also involves RNAi. We found that multiple sRNAs are generated from the mRNA of CD95L within cells and are loaded into the RNA-induced Silencing Complex (RISC), the key mediator of RNAi (*Liu et al., 2004*). Furthermore, we provide evidence that endogenous mRNAs can be processed into sRNAs and loaded into the

RISC, especially in cells with low expression of miRNAs. The subset of genes found in the RISC are regulators of protein translation and cell proliferation.

## Results

### CD95L mRNA is toxic to cells

By testing every possible shRNA derived from the CD95L open reading frame (ORF) or its 3'UTR, we recently found a high enrichment of toxic si/shRNAs derived from the CD95L ORF (*Putzbach et al., 2017*). More recently we determined that the 6mer seed toxicity observed in many si/shRNAs is largely due to their nucleotide composition, with G-rich seeds being the most toxic (*Gao et al., 2018*) (https://6merdb.org). When reanalyzing the CD95L ORF-derived shRNAs, we found a significant correlation between the toxicity of the most toxic CD95L-derived shRNAs (*Putzbach et al., 2017*) and the toxicity of the corresponding 6mer seed we recently determined in a screen of all 4096 6mer seeds (*Gao et al., 2018*). This suggests that CD95L-derived si/shRNAs kill cancer cells, in large part through 6mer seed toxicity.

We therefore wondered whether expression of the CD95L ORF mRNA—without pre-processing into artificial siRNAs—would be toxic to cells. Expression of CD95L protein in most cells kills through the induction of apoptosis. Consequently, expressing CD95L in HeyA8 cells, which are highly sensitive to CD95 mediated apoptosis, killed cells within a few hours after infection with a lentivirus encoding CD95L (*Figure 1A*, left panel). Interestingly, severe growth reduction was seen without any signs of apoptosis (not shown) when a CD95L mutant, unable to bind CD95, was expressed (CD95L$^{MUT}$ in *Figure 1A*, left panel). This mutant carries an Y218R point mutation, that prevents the CD95L protein from binding to CD95 (*Schneider et al., 1997*), and is expressed at a similar level to wild type (wt) CD95L (*Figure 1B*). To prevent the CD95L mRNA from producing full-length CD95L protein, we also introduced a premature stop codon right after the start codon in the CD95L$^{MUT}$ vector (CD95L$^{MUT}$NP). This construct (containing four point mutations and confirmed to produce mRNA with no detectable full-length CD95L protein, *Figure 1B*) was similarly active in reducing the growth of HeyA8 cells compared to the CD95L$^{MUT}$ vector (*Figure 1A*, left panel). CD95L$^{MUT}$NP still produced truncated and likely soluble CD95L (*Figure 1B*), likely due to use of an alternative start codon. However, the supernatant of 293T cells infected with this virus did not induce apoptosis when added to HeyA8 cells (data not shown) confirming that it had no apoptosis inducing activity.

This result suggested that the CD95L mRNA could be toxic to HeyA8 cells without the CD95L protein inducing apoptosis. This was confirmed by expressing the three CD95L constructs in the presence of the oligo-caspase inhibitor zVAD-fmk (*Figure 1A*, center panel). With suppressed apoptosis, all three constructs were now equally toxic to HeyA8 cells. Finally, we tested a HeyA8 CD95 k.o. clone confirmed to express no CD95 protein (*Putzbach et al., 2017*). In these cells, without the addition of zVAD-fmk, wt CD95L and CD95L$^{MUT}$NP were again equally active in reducing the growth of the cells (*Figure 1A*, right panel). Together, these data suggested it is the CD95L mRNA that killed the cells. Cell death was confirmed by quantifying nuclear fragmentation (*Figure 1C*). We also detected a significant increase of ROS in cells expressing CD95L$^{MUT}$NP (*Figure 1D*), which is a characteristic feature of DISE (*Hadji et al., 2014*; *Patel and Peter, 2018*). To exclude the possibility that truncated CD95 protein or any part of the CD95 mRNA would play a role in this toxicity, we deleted the CD95 gene in MCF-7 cells (*Figure 1—figure supplement 1*). Overexpression of wild-type CD95L killed clone FA4 cells, which harbor a complete homozygous deletion of the entire CD95 gene (exon 1–9), as well as CD95 protein k.o. clone #21 cells that retain some truncated mRNA expression (*Figure 1E*).

Finally, to exclude that the truncated protein produced by the CD95L$^{MUT}$NP construct had any unintended activity, we mutated the ATG start codon that most likely gave rise to the truncated CD95L to ATA (*Figure 1—figure supplement 2A*). This CD95L$^{MUT}$NP (G258A) mutant construct did not produce any detectable protein anymore (*Figure 1—figure supplement 2B*) but was as toxic to the CD95 k.o. HeyA8 cells as CD95L$^{MUT}$NP (*Figure 1—figure supplement 2C*).

Our experiments demonstrated that a small number of point mutations introduced into various CD95L mutants did not significantly affect the toxicity of CD95L mRNA. To determine whether a

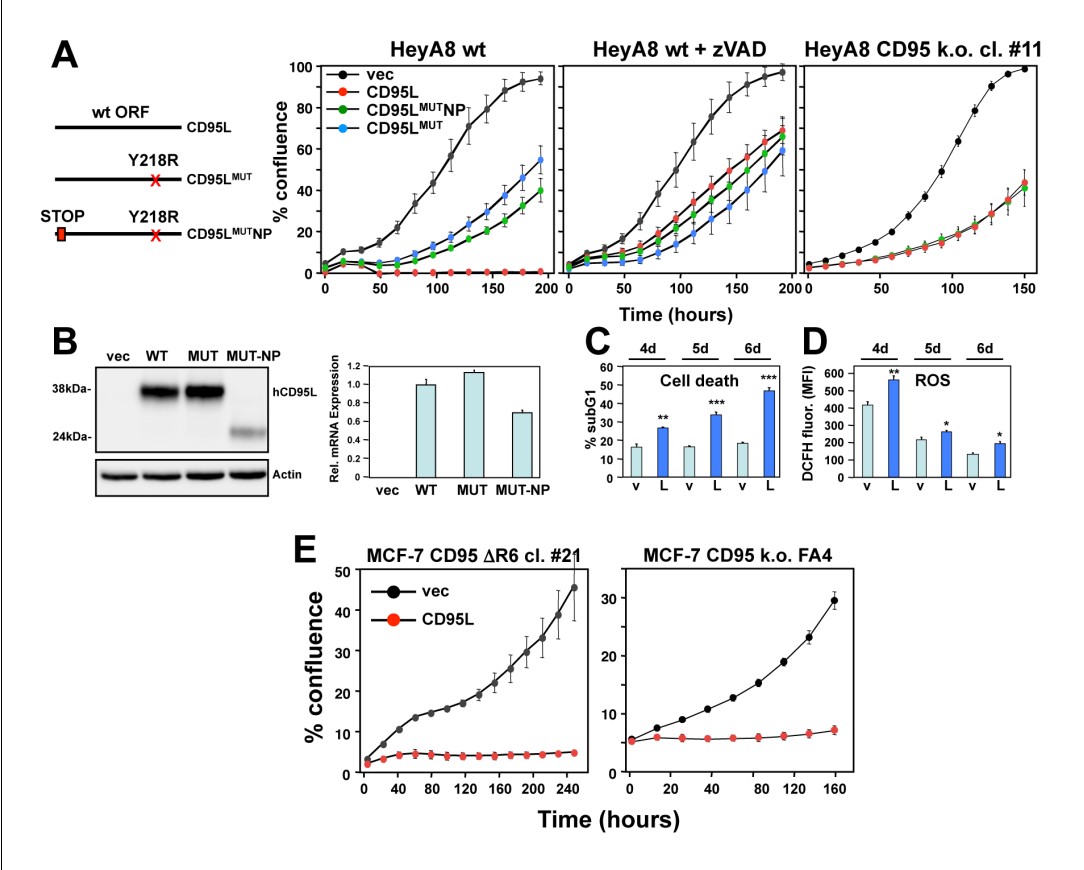

**Figure 1.** The CD95L mRNA is toxic to cells. (A) *Left*: Schematic of the different CD95L mutants used. *Right:* Percent cell confluence over time of HeyA8 parental cells in the absence (*left panel*) or in the presence of 20 μM zVAD-fmk (*center panel*), or CD95 k.o. cells (*right panel*) after expression of CD95L constructs. Data are representative of one to three independent experiments. Values were calculated from samples done in triplicate or quadruplicate shown as mean ±SE. (B) *Left*: Western blot analysis of HeyA8 cells overexpressing different CD95L mutant RNAs. Cells expressing CD95L$^{MUT}$ or CD95L were pretreated with 20 μM zVAD-fmk. Note the small amount of truncated CD95L in cells infected with CD95L $^{MUT}$NP does not have CD95 binding activity. Very similar data were obtained when the constructs were expressed in either CD95 k.o. HeyA8 cells (clone #11) or NB-7 cells, which lack expression of caspase-8, both without treatment with zVAD (data not shown). *Right*: RT-qPCR analysis for CD95L of the same samples. Data are representative of two independent experiments. Each bar represents mean ±S.D. of three replicates. (C, D) Quantification of cell death (C) and ROS production (D) in CD95 k.o. HeyA8 cells (clone #11) expressing either pLenti (v) or pLenti-CD95L (L) at different time points (days after infection). Data are representative of two independent experiments. Each bar represents mean ±SE of three replicates. *p<0.05, **p<0.001, ***p<0.0001, unpaired t-test. (E) Confluency over time of the MCF-7 complete CD95 k.o. FA4 clone (right) or a MCF-7 clone #21 in which we deleted the shR6 target site resulting in an out-of-frame shift after infection with either pLenti vector control (vec) or wt CD95L. Data are representative of two independent experiments. Each data point represents mean ±SE of three replicates.

DOI: https://doi.org/10.7554/eLife.38621.002

The following figure supplements are available for figure 1:

**Figure supplement 1.** Generation of complete CD95 k.o. MCF-7 cells.
DOI: https://doi.org/10.7554/eLife.38621.003
**Figure supplement 2.** Mutation of the alternative start codon in CD95L$^{MUT}$NP construct.
DOI: https://doi.org/10.7554/eLife.38621.004
**Figure supplement 3.** Toxicity of CD95L mRNA is independent of CD95L protein expression.
DOI: https://doi.org/10.7554/eLife.38621.005

large number of mutations would reduce its toxicity, we generated a CD95L$^{MUT}$NP mutant with 308 codon optimized silent point mutations (CD95L$^{SIL}$) (*Figure 1—figure supplement 3A*). The activity of this mutant construct to negatively affect cell growth of CD95 k.o. HeyA8 cells was compared to two independently cloned wt CD95L constructs (WT1 and WT2 in *Figure 1—figure supplement 3B*). The mutant SIL construct was equally effective in suppressing cell growth and produced about the same amount of CD95L mRNA (*Figure 1—figure supplement 3C*). However, the SIL construct

only produced about 12% of WT CD95L protein (*Figure 1—figure supplement 3C*), again supporting the observation that it is the CD95L mRNA and not the protein that elicits toxicity.

We recently determined that a general toxicity of si/shRNAs is largely based on the composition of the 6mer seed sequence of the guide strand (*Gao et al., 2018*). We now plotted the toxicity of all sRNAs that can be derived from either the WT or the SIL mutant CD95L and found that the overall predicted toxicity based on our toxicity screen performed in HeyA8 cells was not statistically different between the two CD95L constructs, (*Figure 1—figure supplement 3D*) suggesting that toxic sequences are embedded in the CD95L$^{SIL}$ ORF. Taken together, our data indicate that even mutating more than 300 positions in CD95L mRNA still allows it to form toxic sRNAs suggesting that the toxicity of CD95L mRNA is a highly robust phenomenon.

## CD95L mRNA kills cells through DISE

After infection with CD95L, CD95 k.o. HeyA8 cells exhibited morphological changes strikingly similar to the changes seen in wt HeyA8 cells after introduction of a CD95L-derived shRNA (shL3) (*Figure 2A*, *Videos 1–4*) suggesting the cells died through a similar mechanism. To determine the cause of cell death induced by CD95L mRNA in HeyA8 CD95 k.o. cells molecularly, we performed a RNA-Seq analysis. We found that expression of CD95L caused preferential downregulation of critical survival genes and not of nonsurvival genes in a control set (*Figure 2B*). In addition, cell death induced by CD95L mRNA resulted in a substantial loss of 11 of the 12 histones detected to be downregulated in cells treated with CD95 and CD95L-derived sh/siRNAs (*Figure 2C*). Loss of histones is an early event during DISE (*Putzbach et al., 2017*). A Metascape analysis demonstrated that nucleosome assembly, regulation of mitosis, and genes consistent with the involvement of histones were among the most significantly downregulated RNAs across all cells in which DISE was induced by any of the four sh/siRNAs or by the expression of CD95L mRNA (*Figure 2D*). This suggests that CD95L mRNA kills cells in the same way as CD95/L-derived si/shRNAs.

## CD95L mRNA kills cells through RNAi

Given our previous work on CD95L-derived si/shRNA toxicity, we hypothesized that CD95L mRNA kills cells through an RNAi-based mechanism—perhaps by being processed into sRNAs that are incorporated into the RISC. Drosha k.o. cells lacking the majority of endogenous miRNAs, but retaining expression of Ago proteins, were shown to be hypersensitive to DISE induced by si- and shRNAs (*Putzbach et al., 2017*). We interpreted this effect as being caused by an increased pool of unoccupied RNAi machinery due to the absence of most miRNAs. Drosha k.o. cells were also hypersensitive to the expression of CD95L$^{MUT}$NP (*Figure 3A*, p=0.014, according to a polynomial fitting model); Virtually all cells died (insert in *Figure 3A*). To directly determine whether the RISC is involved in the toxicity, we introduced CD95L into CD95 k.o. HeyA8 cells after knocking down AGO2 (*Figure 3B*). Toxicity elicited by CD95L was blocked following AGO2 knockdown, suggesting that AGO2 was required for CD95L mRNA to be toxic. This was also the case when CD95 k.o. HeyA8 cells were infected with either the CD95$^{MUT}$NP or the CD95$^{SIL}$ construct (*Figure 3—figure supplement 1A*), confirming that all three CD95L versions killed cells with the help of the RISC. When parental HeyA8 cells were infected with CD95L WT or CD95L$^{SIL}$ they died so rapidly that the cells had to be plated before treatment with puromycin used to select for virus infected cells (*Figure 3—figure supplement 1B*). As expected in these cells that died of canonical apoptosis, knockdown of AGO2 did not have any effect on cell viability. In contrast, cell death induced by CD95L$^{MUT}$NP which in this experiment very efficiently killed the parental HeyA8 cells was severely blunted upon AGO2 knockdown (*Figure 3—figure supplement 1C*).

To test the hypothesis that Drosha k.o. cells were more sensitive because their RISC was not occupied by large amounts of nontoxic miRNAs and to determine whether CD95L mRNA could give rise to sRNAs that are incorporated into the RISC, we pulled down AGO1-4-associated RNAs and analyzed their composition in wt and Drosha k.o. cells after expressing the CD95L$^{MUT}$NP mRNA. For the pull-down, we used a peptide derived from the Ago-binding protein GW182/TNRC6B recently described to bind to all four Ago proteins (*Hauptmann et al., 2015*). As expected in wt HCT116 cells, large amounts of sRNAs (19-23nt in length) were detected bound to the Ago proteins (*Figure 3C*). Both AGO1 and AGO2 were efficiently pulled down. In contrast, in the Drosha k.o. cells, which cannot generate canonical miRNAs, only a low amount of sRNAs was detected,

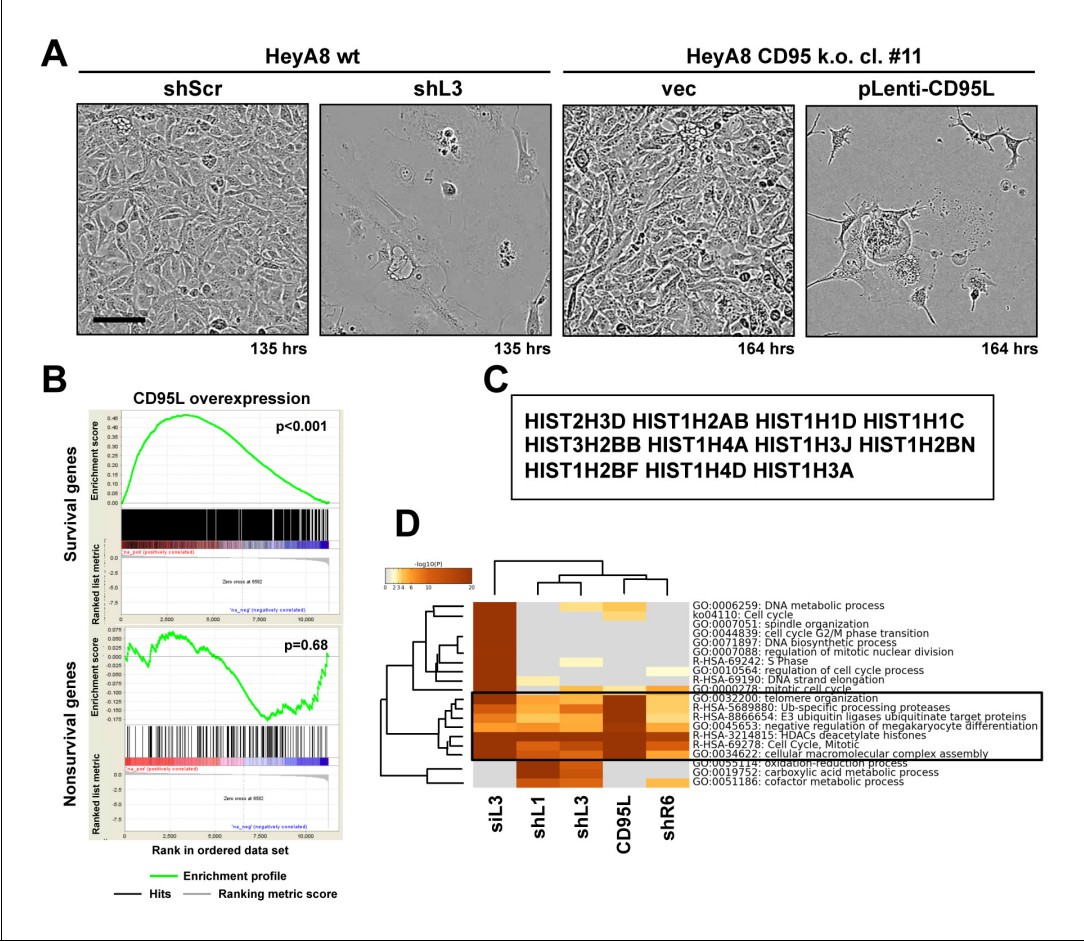

**Figure 2.** Toxicity induced by CD95L overexpression is reminiscent of DISE. (**A**) Phase-contrast images of HeyA8 and HeyA8 CD95 k.o. cells (cl. #11) after infection with pLKO-shScr/shL3 or pLenti (vec)/pLenti-CD95L, respectively, at the indicated time point. (**B**) Gene set enrichment analysis for the 1846 survival genes (*top panel*) and the 416 nonsurvival genes (*bottom panel*) identified in the Sabatini study (*Putzbach et al., 2017*; *Wang et al., 2015*) of mRNAs downregulated in CD95L expressing HeyA8 CD95 k.o. cells compared to HeyA8 CD95 k.o. cells infected with pLenti virus. p-values indicate the significance of enrichment. (**C**) Common genes downregulated in all RNA-Seq experiments from (HeyA8) cells treated with either one of four si/shRNAs (*Putzbach et al., 2017*) derived from either CD95 or CD95L (see *Figure 2D*) and cells overexpressing CD95L ORF as described in (**B**). (**D**) Metascape analysis of 5 RNA Seq data sets analyzed. The boxed GO term clusters were highly enriched in all five data sets.
DOI: https://doi.org/10.7554/eLife.38621.006

The following videos are available for figure 2:

**Figure 2—video 1.** CD95 k.o. HeyA8 cells (cl. #11) infected with pLenti control virus.
DOI: https://doi.org/10.7554/eLife.38621.007

**Figure 2—video 2.** CD95 k.o. HeyA8 cells (cl. #11) infected with pLenti-CD95L virus.
DOI: https://doi.org/10.7554/eLife.38621.008

**Figure 2—video 3.** HeyA8 cells infected with pLKO-shScr.
DOI: https://doi.org/10.7554/eLife.38621.009

**Figure 2—video 4.** HeyA8 cells infected with pLKO-shL3.
DOI: https://doi.org/10.7554/eLife.38621.010

confirming the absence of miRNAs in the RISC. Surprisingly, the amount of pulled down Ago proteins was severely reduced despite the fact these Drosha k.o. cells express comparable levels of AGO2 (*Putzbach et al., 2017*). This suggests the peptide did not have access to the Ago proteins in Drosha k.o. cells, presumably because it only efficiently binds to Ago proteins complexed with RNA as recently shown (*Elkayam et al., 2017*).

The analysis of all Ago-bound RNAs showed that in the wt cells, >98.4% of bound RNAs were miRNAs. In contrast, only 34% of bound RNAs were miRNAs in Drosha k.o. cells (*Figure 3D* and

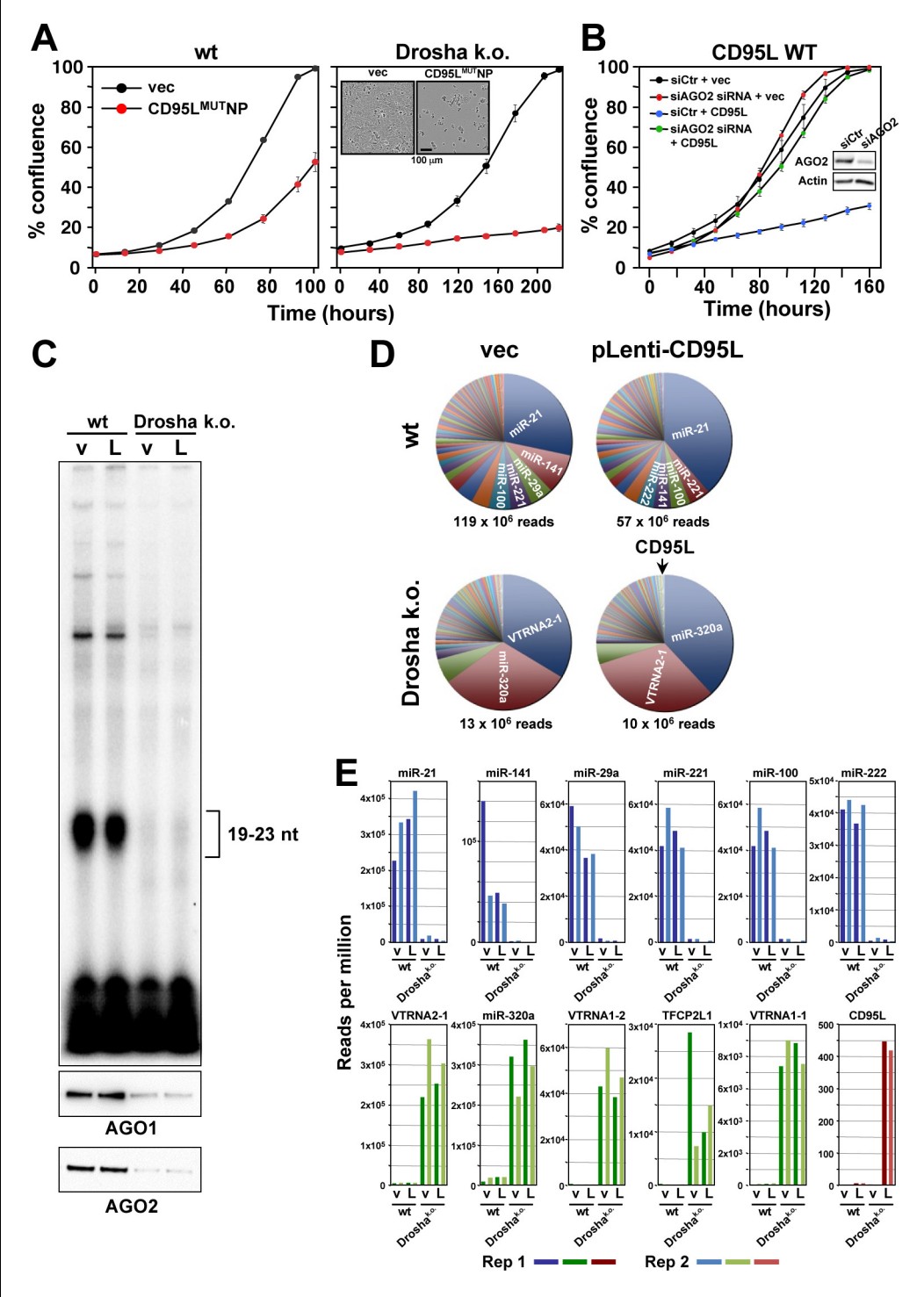

**Figure 3.** Small RNAs generated in cells expressing CD95L mRNA are loaded into the RISC. (**A**) Percent cell confluence over time of HCT116 parental (*left*) or Drosha k.o. (*right*) cells after infection with CD95$^{MUT}$NP. Data are representative of three independent experiments. Each data point represents the mean ±SE of three replicates. *Inset:* Phase contrast images of Drosha k.o. cells 9 days after infection with either empty vector or CD95L$^{MUT}$NP. (**B**) Percent cell confluence of HeyA8 CD95 k.o. cells transfected with either non-targeting siRNA (siCtr) or a pool of 4 siRNAs targeting AGO2 following subsequent infection with either empty pLenti (vec) or pLenti CD95L. *Inset:* Western blot showing knock-down of human AGO2. (**C**) *Top:* autoradiograph on RNAs pulled down with the Ago binding peptide. *Bottom:* Western blot analysis of pulled down Ago proteins. v, pLenti; L, pLenti-CD95L

*Figure 3 continued on next page*

*Figure 3 continued*

expressing cells. (**D**) Pie charts showing the relative ratio of sRNAs pulled down with the Ago proteins in wt and Drosha k.o. cells. Depicted are all the amounts of all sRNAs that contributed at least 0.01% to the total RNA content. Only in the Drosha k.o. cells was a significant amount of CD95L derived Ago bound reads found. They represented the 75th most abundant sRNA species (arrow). The average number of total sequenced reads (of two duplicates) are shown for each condition. (**E**) *Top*: Number of reads (normalized per million) of the top six most abundant sRNAs in the RISC of either HCT116 wt-pLenti or -pLenti-CD95L cells. *Bottom*: Number of reads (per million) of the top five genes with sRNAs most abundant in the RISC or of CD95L in the RISC of either HCT116 Drosha k.o. pLenti (v), or -pLenti-CD95L (L) cells. Note: miR-21 is not included as it is already shown in the top row. Bottom right panel: Abundance of Ago bound CD95L derived sRNAs. Shown in all panels is the abundance of RNAs in the four samples. Rep 1 and Rep 2, replicate 1 and 2.
DOI: https://doi.org/10.7554/eLife.38621.011
The following figure supplement is available for figure 3:

**Figure supplement 1.** All CD95L mRNA mutants are toxic through RNAi.
DOI: https://doi.org/10.7554/eLife.38621.012

data not shown). These include miRNAs that are processed independently of Drosha such as miR-320a (*Kim et al., 2016*). Consequently, this miRNA became a major RNA species bound to Ago proteins in Drosha k.o. cells (*Figure 3D*). In both wt and Drosha k.o. cells, a significant increase in CD95L-derived sRNAs bound to the Ago proteins was detected in cells infected with the CD95L virus compared to cells infected with pLenti empty vector, as expected. They corresponded to 0.0006% and 0.043% of all the Ago-bound RNAs in the wt cells and Drosha k.o. cells, respectively. Toxicity of CD95L mRNA was, therefore, not due to overloading the RISC. In the absence of most miRNAs, the total amount of RNAs bound to Ago proteins in the Drosha k.o. cells was roughly 10% of the amount bound to Ago in wt cells (*Figure 3D*). The reduction of Ago-bound miRNAs in Drosha k.o. cells (*Figure 3E*, top row) was paralleled by a substantial increase in binding of other sRNAs to the Ago proteins (*Figure 3E*, bottom row). Interestingly, the amount of Ago-bound CD95L-derived sRNAs was >100 times higher in the Drosha k.o. cells compared to the wt cells (red columns in *Figure 3E*). These data support our hypothesis that Drosha k.o. cells are more sensitive to CD95L mRNA-mediated toxicity due to their ability to take up more toxic small CD95L-derived RNAs into the RISC in the absence of most miRNAs.

## CD95L ORF is degraded into sRNA fragments that are then loaded into the RISC

Interestingly, not only did Ago proteins in Drosha k.o. cells bind much more CD95L-derived sRNAs than in the wt cells, but also the peak length of the most abundant Ago-bound RNA species increased from 20 to 23 nt (*Figure 4A*, top panel). To determine the sites within the CD95L mRNA that gave rise to small Ago-bound RNAs, we aligned all small Ago-bound RNAs detected in all conditions to the CD95L ORF sequence (*Figure 4B and C*). We identified 22 regions in the CD95L ORF that gave rise to sRNAs that were bound by Ago proteins (*Figure 4B*). To determine whether these sRNAs were formed in the cytosol and then loaded into the RISC, we also aligned all sRNAs in the total RNA fraction isolated from CD95L$^{MUT}$NP expressing HCT116 Drosha k.o. cells with CD95L (*Figure 4C*). Interestingly, very similar regions of small RNAs were found. Moreover, the mean as well as the peak of the distribution of the read lengths of sRNAs bound to Ago proteins was smaller than in the total small RNAs fraction (*Figure 4A*, center panel), suggesting these fragments were trimmed to the appropriate length either right before they were loaded into the RISC or by the RISC itself. This was most obvious for the sRNAs in cluster 3 (*Figure 4B and C*). We also noticed that certain sRNAs were more abundant in the Ago-bound fraction when compared to total RNA relative to all other RNAs. To determine whether this type of processing was specific for HCT116 Drosha k.o. cells, we analyzed the Ago-bound CD95L-derived sRNAs in HeyA8 CD95 k.o. cells after expression of wt CD95L (*Figure 4D*) and compared them with the total RNA fraction (*Figure 4E*). While we found fewer CD95L-derived reads in these cells, the general location of some of the read clusters overlapped with the ones found in the Drosha k.o. cells and again both the mean and peak of the distribution of RNA lengths was smaller in the Ago-bound fraction versus the total RNA fraction (*Figure 4A*, bottom panel). Together, these data suggest that CD95L mRNA can be processed into

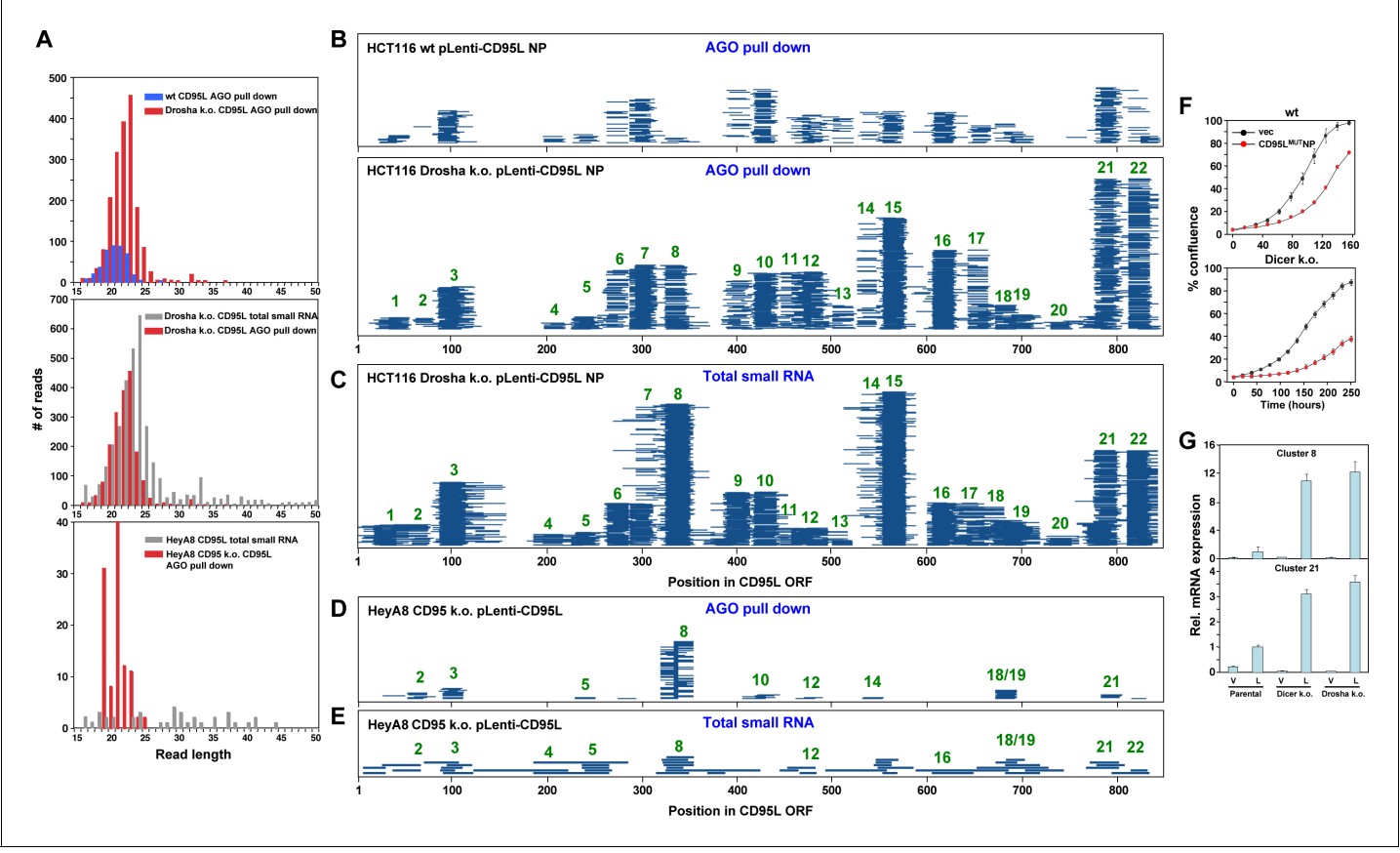

**Figure 4.** The entire CD95L mRNA gives rise to sRNAs that bind to the RISC. (A) Length distribution of CD95L derived reads in various analyses. (B, C) Read alignment with CD95L$^{MUT}$NP ORF of analyses of sRNAs pulled down with Ago proteins from HCT116 wt (B, top) and Drosha k.o. (B, bottom) cells and of total sRNAs from HCT116 Drosha k.o. cells (C) after infection with CD95L$^{MUT}$NP. (D, E) Read alignment with wt CD95L ORF of analyses of sRNAs pulled down with Ago proteins (D) or total sRNAs (E) from HeyA8 CD95 k.o. cells after infection with wt CD95L. (F) Percent cell confluence over time of HCT116 parental (*top*) or Dicer k.o. (clone #43) (*bottom*) cells after infection with CD95$^{MUT}$NP. (Dicer k.o. clone #45, gave a similar result, data not shown). Data are representative of two independent experiments. Each data point represents the mean ±SE of three replicates. (G) RT-qPCR analysis of clusters 8 and 21 in HCT116 parental, Dicer k.o. (clone #43), and Drosha k.o. cells after infection with CD95$^{MUT}$NP. Each bar represents mean ± S.D. of three replicates. v, vector, L, CD95L expressing cells.

DOI: https://doi.org/10.7554/eLife.38621.013

The following figure supplements are available for figure 4:

**Figure supplement 1.** Predicted secondary structure of CD95L ORF and toxicity of CD95L-derived sRNAs after conversion to siRNAs.

DOI: https://doi.org/10.7554/eLife.38621.014

**Figure supplement 2.** CD95L fragments are less toxic than full length CD95L mRNA.

DOI: https://doi.org/10.7554/eLife.38621.015

smaller RNA fragments, which are then trimmed to a length appropriate for incorporation into the RISC.

Our data suggest that the CD95L mRNA, when overexpressed, is toxic to cells due to the formation of Ago-bound sRNAs that are incorporated into the RISC and kill cells through RNAi. This process is independent of Drosha. To determine whether Dicer is required for either processing of CD95L mRNA or loading the sRNAs into the RISC, we expressed CD95L$^{MUT}$NP in wt and Dicer k.o. HCT116 cells (*Figure 4F*). Dicer k.o. cells were still sensitive to toxicity induced by CD95L mRNA expression, suggesting the toxicity of the CD95L mRNA does not require the processing by either Drosha or Dicer. Using custom real-time qPCR primers designed to specifically detect the sRNAs from clusters 8 and 21, we detected, in both wt and Dicer k.o. cells over-expressing CD95L$^{MUT}$NP, fragments from these clusters (*Figure 4G*), demonstrating that Dicer is not involved in processing CD95L mRNA.

All the reported small RNAs derived from CD95L corresponded to the sense strand of the expressed mRNA, raising the question of how they could be processed into double-stranded sRNAs in the absence of an antisense strand. To get a preliminary answer to this question, we subjected the CD95L ORF mRNA sequence to a secondary structure prediction (*Figure 4—figure supplement 1A*). According to this analysis, the CD95L ORF mRNA forms a tightly folded structure with many of the sRNAs of the 22 clusters juxtaposing each other in stem-like structures creating regions of significant complementarity. These may provide the duplexes needed to be processed and loaded into the RISC. Interestingly, some of the juxtaposing reads were found in duplex structures with 3' overhangs. Three of these oligonucleotides (derived from clusters 7, 15 and 22) when expressed as siRNAs were toxic to HeyA8, H460, M565 and 3LL cells (*Figure 4—figure supplement 1B*).

To address the question of whether the complete CD95L ORF sequence was required to produce toxic sRNAs, we generated two CD95L fragments, one 5' fragment (280 nt) and one 3' fragment (559 nt) (*Figure 4—figure supplement 2A*). When expressed in CD95 k.o. HeyA8 cells, no protein was detected with the anti-CD95L antibody used but they were both significantly expressed at the RNA level albeit at lower levels than wt CD95L (*Figure 4—figure supplement 2B*). Only the 5' fragment caused a small reduction in cell growth when compared to full length CD95L (*Figure 4—figure supplement 2C*). While neither of the two fragments were toxic to wt HCT116 cells, in the hypersensitive Drosha k.o. HCT116 cells both fragments showed weak toxicity (*Figure 4—figure supplement 2D and E*). These data suggest that fragments of CD95L show reduced toxicity which could be due to their lower expression levels and/or their inability to properly fold and form sRNAs.

To address the question of whether in addition to exogenously expressed CD95L, mRNAs of endogenous genes would also be processed and loaded into the RISC, we interrogated the RNA-Seq data sets we had from Ago pull-down and total small RNA-Seq analyses in HCT116 wt and HCT116 Drosha k.o. cells. Genes processed in a similar way to CD95L mRNA would have to be significantly expressed at the mRNA level, and would give rise to multiple sRNAs that bound to the RISC more effectively in Drosha k.o. cells when compared to wt cells. We interrogated all data sets using the following parameters: We counted any gene that had at least one unique read that aligned only within its mRNA with a read count of 10 or more. We started by analyzing the Ago bound reads and then compared the location and size of the reads in these identified genes with the analysis of total small RNA of Drosha k.o. cells expressing CD95L. We found Ago bound reads of 10 or more reads/location that aligned with 5629 genes in the human genome. Of these genes ~ 10% were non-coding, 558 (~10%) were protein coding genes that met our criteria for processing, and 4498 genes (~80%) were protein coding genes that did not. The top ten most abundant processed coding genes with multiple unique reads in the RISC are shown in *Figure 5A and B* and *Figure 5—figure supplement 1A and B*. The ones most evenly processed are shown in *Figure 5B*. In almost all cases, similar to processed CD95L, the average length of the reads bound to the Ago proteins was smaller than the ones found in total RNA (*Figure 5A* and *Figure 5—figure supplement 1A*). While these genes were abundantly expressed, they were not the most abundant mRNAs in these cells (*Figure 5C*). Of the protein coding genes with reads bound to Ago proteins many of them were highly expressed, yet they were not processed in this form (*Figure 5D*). When all small reads derived from these five genes were aligned with the human genome, we observed that: 1) The reads were between 5–10 times more abundant in the RISC of cells devoid of Drosha expression and 2) they were loaded equally efficiently in cells expressing the empty vector control or the CD95L expressing virus (*Figure 5—figure supplement 2*).

The fact that many highly expressed protein coding genes were not processed raised the question of whether the processed and RISC loaded genes fell in certain functional clusters. To begin to address this question, we subjected all processed coding genes with RISC bound sRNAs to a DAVID gene ontology analysis. Strikingly, genes found in many of these enriched clusters were involved in cell growth (green rows in *Figure 5—figure supplement 3A*). In fact, the most significantly enriched genes were involved in protein translation (dark green rows in *Figure 5—figure supplement 3A*). Interestingly, in addition to cyclin D1, MYC (*Figure 5B*) and CDK6 (*Supplementary File 1*) and many initiation and elongation factors, 38 of the mRNAs of the 50 ribosomal proteins (76%) in the analysis were processed in this way (*Figure 5—figure supplement 3B*). This suggests that proteins involved in protein translation may be marked to be processed and loaded into the RISC.

Our data suggest that mRNAs of endogenous protein coding genes can be processed and loaded into the RISC especially under low miRNA conditions. This raised the question of whether

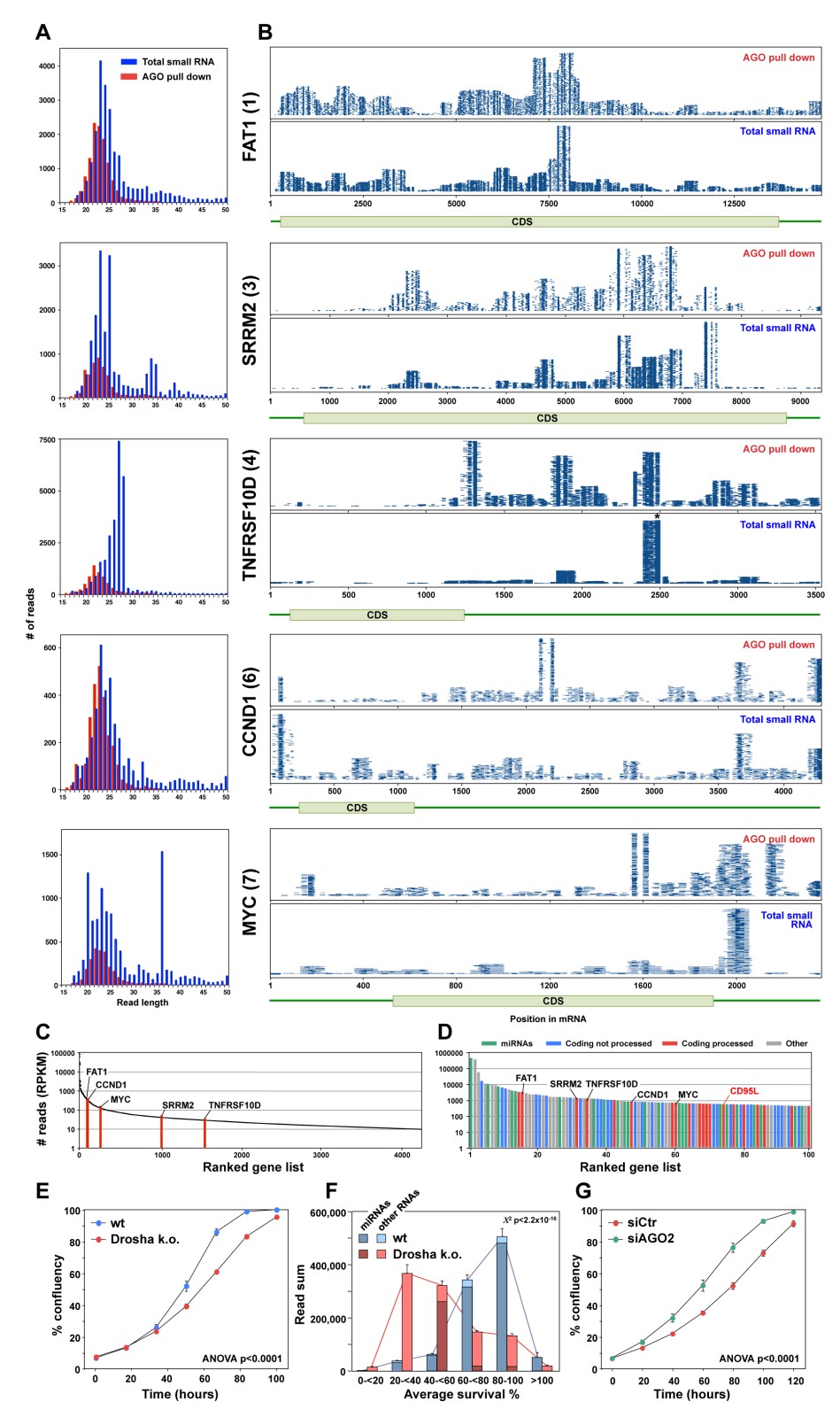

**Figure 5.** Endogenous mRNAs are processed and loaded into the RISC. (**A**) Length distribution of reads derived from five of the top ten most abundant genes loaded into the RISC of CD95L expressing HCT116 Drosha k.o. cells. The numbers in parentheses indicate the ranking in the top ten most abundant genes with Ago bound reads. (**B**) Alignment of the reads from the five genes shown in A with horizontal blue lines representing the mapped positions of the reads. Each blue line represents an individual read, with its length in the plot proportional to the read length. Small RNAs
*Figure 5 continued on next page*

*Figure 5 continued*

pulled down with Ago proteins (top) or total sRNAs (bottom) from HCT116 and Drosha k.o. cells after infection with wt CD95L. *This stack contains 14899 reads of which 3000 were randomly chosen and plotted. (C) All 4262 genes in HCT116 Drosha k.o. cells expressing CD95L ranked according to highest expression with more than 10 reads expressed as reads per kb per million (RPKM). The abundance of the six genes shown in A and B is labeled. (D) Genes ranked according to highest abundance in the RISC of Drosha k.o. cells. Reads derived from the five genes in A are labeled as well as the location of the reads derived from CD95L. (E) Percent cell confluence over time of parental HCT116 and Drosha k.o. cells. (F) Average seed toxicity of all Ago-bound miRNAs and non-miRNAs (Other) in parental HCT116 and Drosha k.o. cells. Reads are shown as reads per million (RPM). Chi squared test was used to calculate p-value. (G) Percent cell confluence over time of Drosha k.o. HCT116 cells 24 hr after transfection with 25 nM of either nontargeting SMARTpool (siCtr) or AGO2 SMARTpool siRNAs. Each data point represents mean ±SE of three replicates. The experiment is representative of three biological repeats.

DOI: https://doi.org/10.7554/eLife.38621.016

The following figure supplements are available for figure 5:

**Figure supplement 1.** Endogenous mRNAs are processed and loaded into the RISC - additional genes.
DOI: https://doi.org/10.7554/eLife.38621.017
**Figure supplement 2.** Mapping of Ago bound reads from five processed genes to the human genome.
DOI: https://doi.org/10.7554/eLife.38621.018
**Figure supplement 3.** Genes with multiple reads bound to Ago proteins are involved in cell growth and protein translation.
DOI: https://doi.org/10.7554/eLife.38621.019

RISC loaded sRNAs could negatively regulate cell growth. We noticed that Drosha k.o. cells consistently grew slower than their parental wt cells (*Figure 5E*). Based on our recent data on 6mer seed toxicity we wondered whether in the absence of most nontoxic and therefore protective miRNAs cells would incorporate sRNAs into the RISC that negatively affect cell growth through a low level of survival gene targeting. To address this question, we compared the 6mer seed toxicity of all sRNAs bound to Ago proteins in either wt or Drosha k.o. cells (*Figure 5F*). The difference in predicted seed toxicity was striking. Most RISC bound sRNAs in wt cells were miRNAs with low seed toxicity (blue columns in *Figure 5F*). In contrast, in Drosha k.o. cells the majority of RISC bound sRNAs had a rather toxic 6mer seed (red columns in *Figure 5F*). This suggested that such sRNAs may constantly put pressure on the expression of survival genes. To test this, we knocked down AGO2 in the Drosha k.o. cells and this treatment resulted in an increase of their growth rate that was now similar to the one observed in wt cells (*Figure 5G*). These data suggest that endogenous RISC bound sRNAs may regulate cell growth through the DISE mechanism.

In summary, our data suggest that si- and/or shRNAs with certain seed sequences present in CD95 and CD95L and the entire CD95L ORF are toxic to cancer cells. The CD95L mRNA is processed into sRNAs that are loaded into the RISC that then target critical survival genes. This results in cell death largely through 6mer seed toxicity. This process is independent of both Drosha and Dicer. Furthermore, our data suggest that a high miRNA content, by 'filling up' the RISC, might render cells less sensitive to this form of cell death and that multiple endogenous mRNAs are processed and loaded into the RISC and may regulate cell fate.

## Discussion

We recently reported a novel form of cell death that was observed after expression of si/shRNAs designed from the sequences of CD95/CD95L mRNA (*Putzbach et al., 2017*). More recently we described that cells die from a loss of multiple survival genes through a mechanism we call 6mer seed toxicity (*Gao et al., 2018*). The most toxic si/shRNAs derived from CD95 or CD95L were found in the ORF of CD95L (*Putzbach et al., 2017*). This pointed toward the CD95L mRNA itself being toxic.

We now show that expression of full-length CD95L mRNA triggers toxicity that is independent of the protein product and canonical apoptosis. This is intriguing considering that previous studies showed transgenic expression of CD95L using viruses killed multiple cancer cells that were completely resistant to CD95 mediated apoptosis (*ElOjeimy et al., 2006*; *Hyer et al., 2000*; *Sudarshan et al., 2005*; *Sun et al., 2012*). These results were interpreted as intracellular CD95L triggering apoptosis more efficiently than when added to the cells. However, we now provide an alternate explanation—namely, both the CD95L protein *and* mRNA are toxic to cells through distinct

mechanisms. The protein induces apoptosis, and the mRNA induces toxicity through an RNAi-based mechanism.

We demonstrate that Dicer and Drosha are not involved in generating the Ago-bound CD95L-derived fragments but there are several candidate RNases that are capable of processing mRNAs. Given the differences in length distribution between the cytosolic versus Ago-bound RNA fragments, it is likely that CD95L-derived fragment intermediates are incorporated into the RISC and then trimmed to the appropriate length by Ago. Indeed, a similar mechanism is known to occur during the maturation of the erythropoietic miR-451, where the pre-miRNA is first cleaved by AGO2 and then trimmed at the 3' end to the final mature form by the exoribonuclease PARN (*Yoda et al., 2013*). Furthermore, a similar process occurs with the recently identified class of Ago-bound RNAs called agotrons (*Hansen et al., 2016*), which consist of an excised intron loaded into the RISC in a manner independent of Drosha or Dicer pre-processing. Once trimmed to the appropriate size, the guide RNAs in complex with the RISC can regulate gene expression through RNAi.

Our data provide the first evidence of an overexpressed cDNA exerting toxicity via an RNAi-dependent mechanism. It was first shown in plants that overexpressed transgenes can be converted into RNAi active short RNA sequences (*Hamilton and Baulcombe, 1999*). Our data on the effects of overexpressed CD95L RNA, while mechanistically distinct from what was reported in plants, may be the first example of a transgene determining cell fate through the RNAi mechanism in mammalian cells. The CD95L-derived sRNAs will likely act in a miRNA-like fashion by targeting 3'UTRs of survival genes through 6mer seed toxicity (*Gao et al., 2018*). CAG-repeat-containing mRNAs have been shown to induce sRNA formation and cellular toxicity via RNAi (*Bañez-Coronel et al., 2012*). However, we recently reported that these sCAGs likely target fully complementary CUG containing repeat regions in the ORFs of genes critical for cell survival in an siRNA-like mechanism (*Murmann et al., 2018a*; *Murmann et al., 2018b*).

In addition to the activity of exogenously added CD95L mRNA, we also provide evidence that certain endogenous coding mRNAs can be processed into multiple sRNAs that are then loaded into the RISC. Small mRNA-derived RNAs have been reported to be bound to all four Ago proteins before (*Burroughs et al., 2011*). However, they were very small in numbers and amount and no specific cellular function could be ascribed to them. In contrast, we now show that in cells with disabled miRNA processing, about 3% of the the protein coding genes can be processed in this way and that these genes are strongly enriched in GO terms associated with protein translation (initiation and elongation) as well as with cell cycle regulation and cell proliferation. They contain a large number of 5' TOP genes (i.e. 76% of all ribosomal proteins) required for protein translation (*Yamashita et al., 2008*). Our findings may be relevant for situations in which cellular miRNA levels are low, such as in somatic stem cells (*Morin et al., 2008*) or in advanced cancers which are characterized by a global downregulation of miRNAs (*Lu et al., 2005*). It is intriguing to note that the genes that are part of this group fall into GO clusters overlapping with the ones we found downregulated in cells undergoing DISE (*Gao et al., 2018*; *Putzbach et al., 2017*).

A major question that arises from our data is whether CD95L mRNA is toxic in vivo. We and others have noticed upregulation of CD95L in multiple stress-related conditions such as after treatment with chemotherapy ((*Friesen et al., 1999*) and data not shown). While the amount of CD95L mRNA and the level of upregulation alone may not be enough to be toxic, it could be the combination of multiple RNA fragments, derived from multiple different mRNAs that are generated to kill cells (*Putzbach et al., 2018*). We view CD95L as just one of many RNAs that have this activity. It is unlikely CD95L is the only gene whose mRNA is toxic to cells, as this mRNA-based level of toxicity would be redundant with the potent killing capacity of the CD95L protein. Also, upregulating an mRNA that, by itself, could decimate the cells that would otherwise need to upregulate that mRNA to carry out their biological function in the first place, such as in activated T cells upregulating CD95L to mount an immune response, would be self-defeating. Therefore, nature likely distributed this mRNA-based toxicity-inducing capacity over many genes in the genome to prevent activating it when any one of those genes is upregulated during specific cellular processes. It is more likely there exists an entire network of these genes that can release toxic sRNAs when the appropriate stimulus is encountered. Consistent with this hypothesis we recently identified other genes that contain sequences that when converted to shRNAs kill cancer cells through 6mer seed toxicity (*Patel and Peter, 2018*). Future work will be aimed at studying the coding genes that are found in the RISC,

identifying additional genes and the mechanism through which they are processed and under what conditions they kill cells.

# Materials and methods

## Key resources table

| Reagent type (species) or resource | Designation | Source or reference | Identifiers | Additional information |
|---|---|---|---|---|
| Gene (Homo sapiens) | CD95L | NA | NM_000639 | |
| Gene (H. sapiens) | CD95 | NA | NM_000043 | |
| Cell line (H. sapiens) | MCF-7 | ATCC | ATCC: HTB-22 | Human adenocarcinoma of the mammary gland, breast; derived from metastatic site: pleural effusion |
| Cell line (H. sapiens) | MCF-7 CD95 ΔshR6 clone #21 | this paper | NA | MCF-7 CD95 ΔshR6 clone #21 with homozygous 227 nucleotide deletion of the shR6 target site in CD95 (chr10:89,008,920–89,009,146; Human Dec. 2013 GRCh38/hg38 assembly) produced using CRISPR/Cas9 technology; verified homozygous CD95 protein knockout |
| Cell line (H. sapiens) | MCF-7 CD95 deletion clone FA4 | this paper | NA | MCF-7 CD95 deletion clone FA4 with a homozygous deletion of the entire CD95 gene (chr10:88,990,657–89,015,785; Human Dec. 2013 GRCh38/hg38 assembly) produced using CRISPR/Cas9 technology; verified homozygous CD95 protein knockout |
| Cell line (H. sapiens) | HeyA8 | PMID: 4016745 | RRID: CVCL_8878 | Human high grade ovarian serous adenocarcinoma; derived from parent Hey cells (RRID: CVCL_0297) |
| Cell line (H. sapiens) | HeyA8 shR6 k.o. clone #11, HeyA8 CD95 k.o. | PMID: 29063830 | NA | HeyA8 CD95 k.o. clone with a homozygous 227 nucleotide deletion of the shR6 target site in CD95 (chr10:89,008,920–89,009,146; Human Dec. 2013 GRCh38/hg38 assembly) produced using CRISPR/Cas9 technology; verified homozygous CD95 protein knockout |
| Cell line (H. sapiens) | HCT116 | Korean Collection for Type Cultures (KCTC) | KCTC: cat#HC19023; ATCC: CCL_247 | Human colorectal carcinoma |
| Cell line (H. sapiens) | Drosha$^{-/-}$; Drosha$^{-/-}$ clone #40 | Korean Collection for Type Cultures (KCTC); PMID: 26976605 | KCTC: cat#HC19020 | HCT116 clone #40 with homozygous knockout of Drosha protein; knockout achieved using CRISPR/Cas9 which resulted in a single nucleotide insertion in one allele and a 26 nucleotide deletion in the other |
| Cell line (H. sapiens) | Dicer$^{-/-}$; Dicer$^{-/-}$ clone #43 | Korean Collection for Type Cultures (KCTC); PMID: 26976606 | KCTC: cat#HC19023 | HCT116 clone #43 with homozygous knockout of Dicer protein; knockout achieved using CRISPR/Cas9 which resulted in a three nucleotide insertion and 14 nucleotide deletion in one allele and a 35 nucleotide deletion in the other |
| Cell line (H. sapiens) | Dicer$^{-/-}$; Dicer$^{-/-}$ clone #45 | Korean Collection for Type Cultures (KCTC); PMID: 26976607 | KCTC: cat#HC19024 | HCT116 clone #45 with homozygous knockout of Dicer protein; knockout achieved using CRISPR/Cas9 which resulted in a 53 nucleotide deletion in one allele and a 28 nucleotide deletion in the other |
| Cell line (H. sapiens) | 293T | ATCC | ATCC: CRL-3216 | Derived from HEK293 cells (ATCC: CRL-1573); express large T antigen; used for packaging viruses |
| Cell line (H. sapiens) | H460 | ATCC | ATCC: #HTB-177 | Human lung pleural effusion carcinoma |
| Cell line (Mus musculus) | 3LL | ATCC | ATCC #CRL-1642 | Mouse Lewis lung carcinoma |

*Continued on next page*

*Continued*

| Reagent type (species) or resource | Designation | Source or reference | Identifiers | Additional information |
|---|---|---|---|---|
| Cell line (Mus musculus) | M565 | PMID: 25366259 | NA | Mouse hepatocellular carcinoma isolated from naturally occurring tumor in a floxed CD95 background |
| Antibody | anti-human AGO1 (rabbit monoclonal) | Cell Signaling | Cell Signaling #5053 | 1:2000; for western blot; primary Ab |
| Antibody | anti-human AGO1 (rabbit polyclonal) | Abcam | Abcam #98056 | 1:2000; for western blot; primary Ab |
| Antibody | anti-human AGO2 (rabbit polyclonal) | Abcam | Abcam #32381 | 1:500; for western blot; primary Ab |
| Antibody | Goat anti-rabbit, IgG-HRP | Southern Biotech | Southern Biotech: cat#SB-4030–05 | 1:5000; for western blot; secondary Ab |
| Antibody | Anti-Argonaute-2 antibody (rabbit monoclonal) [EPR10411] | Abcam | Abcam #186733 | 1:1200; for western blot; primary Ab |
| Antibody | Anti-Human CD178 antibody (Mouse IgG1) Clone G247-4 | BD Pharmingen | BD Pharmingen #556387 | 1 µg/ml; for western blot; primary Ab |
| Antibody | Anti-CD95 (rabbit polyclonal, C-20) | Santa Cruz | Santa Cruz #sc-715 (since discontinued) | 1:1000; for western blot; primary Ab |
| Antibody | Goat anti-rabbit, IgG-HRP | Cell Signaling | Cell Signaling: cat#7074 | 1:2000; for western blot; secondary Ab |
| Antibody | Goat anti-mouse; IgG1-HRP | Southern Biotech | Southern BioTech: cat#1070–05 | 1:5000; for western blot; secondary Ab |
| Recombinant protein reagent | LzCD95L | PMID: 14504390 | NA | Leucine zipper tagged CD95L; recombinant protein |
| Chemical compound | CellTiter-Glo | Promega | Promega #G7570 | Detects ATP release as a surrogate for cell death; read-out is fluorescence |
| Chemical compound | propidium iodide | Sigma-Aldrich | Sigma-Aldrich: cat#P4864 | Used for subG1 flow cytometry analysis |
| Chemical compound | puromycin | Sigma-Aldrich | Sigma-Aldrich: cat#P9620 | Used for selection of cells expressing puromycin resistance cassettes |
| Chemical compound | 2′,7′-dichlorodihydro fluorescein diacetate | Thermofisher Scientific | Thermofisher Scientific #D399 | Dye used for detecting ROS production |
| Chemical compound | zVAD-fmk | Sigma-Aldrich | Sigma-Aldrich: cat#V116 | Used at 20 uM; pan caspase inhibitor |
| Recombinant DNA reagent | pLenti-GIII-CMV-RFP-2A-Puro vector; pLenti | ABM Inc | NA | pLenti control empty lentiviral vector; carries an RFP-2a-puromycin resistance cassette |
| Recombinant DNA reagent | pLenti-CD95L | this paper | NA | pLenti-GIII-CMV-RFP-2A-Puro vector that expresses human wild type CD95L cDNA (NM_000639.2); used to express wt human CD95L upon infection with lentiviral particles |
| Recombinant DNA reagent | pLenti-CD95L$^{MUT}$ | this paper | NA | pLenti-GIII-CMV-RFP-2A-Puro vector that expresses human CD95L cDNA (NM_000639.2) with two nucleotide substitutions in codon 218 (TAT - > CGT) resulting in replacement of tyrosine for arginine (Y218R mutation); unable to bind CD95 |

*Continued on next page*

*Continued*

| Reagent type (species) or resource | Designation | Source or reference | Identifiers | Additional information |
|---|---|---|---|---|
| Recombinant DNA reagent | pLenti-CD95L$^{MUT}$NP | this paper | NA | pLenti-GIII-CMV-RFP-2A-Puro vector that expresses human CD95L cDNA (NM_000639.2) with both the Y218R mutation and a single nucleotide substitution at the second codon (CAG - > TAG), resulting in a premature stop codon right after the start codon |
| Recombinant DNA reagent | pLenti-CD95L$^{SIL}$ | this paper | NA | pLenti-GIII-CMV-RFP-2A-Puro vector that expresses human CD95L cDNA (NM_000639.2) with all codons containing synonymous mutations except for select codons in the proline-rich domain to meet IDT synthesis criteria |
| Transfected construct | gRNA scaffold | PMID: 23287722 | IDT: synthesized as gene block | 455 nucleotide CRISPR/Cas9 gRNA scaffold synthesized as a gene block; contains promoter, gRNA scaffold, target sequence, and termination sequence; scaffold transcribes gRNAs that target Cas9 endonuclease to cut at target sites; target sequences consist of 19 nucleotides that are complementary to the target site of choice; co-transfected with Cas9 to catalyze cleavage. |
| Recombinant DNA reagent | pLenti-CD95L$^{MUT}$NP (G258A) | This paper | NA | pLenti-GIII-CMV-RFP-2A-Puro vector that expresses human CD95L cDNA (NM_000639.2) with the Y218R mutation, and two additional single nucleotide substitutions; one at the second codon (CAG - > TAG), resulting in a premature stop codon right after the start codon, and another, G258A, resulting in the replacement of a methionine with an isoleucine, thus removing the alternative translational start site. |
| Transfected construct | pMJ920 Cas9 plasmid | Addgene; PMID: 23386978 | Addgene: cat#42234 | Plasmid that expresses a human codon-optimized Cas9 tagged with GFP and HA; used to express Cas9 for CRISPR-mediated deletions. |
| Chemical compound | Lipofectamine 2000 | ThermoFisher Scientific | ThermoFisher Scientific: cat#11668019 | Transfection reagent |
| Chemical compound | Lipofectamine RNAiMAX | ThermoFisher Scientific | ThermoFisher Scientific: cat#13778150 | Transfection reagent; used for transfection of small RNAs such as siRNAs |
| Commercial assay or kit | StrataClone Blunt PCR Cloning Kit | Agilent Technologies | Agilent Technologies: cat#240207 | Used to blunt-end clone the gRNA scaffolds into the pSC-B plasmid |
| Genetic reagent | Taqman Gene expression master mix | ThermoFisher Scientific | #4369016 | |
| Sequence-based reagent | shR6 flanking Fr primer | IDT | IDT: custom DNA oligo | Fr primer that flanks shR6 site; used to detect 227 nt shR6 deletion; 5'-GGTGTCATGCTGTGACTGTTG-3' |
| Sequence-based reagent | shR6 flanking Rev primer | IDT | IDT: custom DNA oligo | Rev primer that flanks shR6 site; used to detect 227 nt shR6 deletion; 5'-TTTAGCTTAAGTGGCCAGCAA-3' |
| Sequence-based reagent | shR6 internal Rev primer | IDT | IDT: custom DNA oligo | Rev primer that overlaps with the shR6 site; used to detect 227 nt shR6 deletion; 5'-AAGTTGGTTTACATCTGCAC-3' |

*Continued on next page*

*Continued*

| Reagent type (species) or resource | Designation | Source or reference | Identifiers | Additional information |
|---|---|---|---|---|
| Sequence-based reagent | CD95 flanking Fr primer | IDT | IDT: custom DNA oligo | Fr primer that flanks the CD95 gene; used to detect CD95 deletion; 5'-TGTTTAATATAGCTGGGGCTATGC-3' |
| Sequence-based reagent | CD95 flanking Rev primer | IDT | IDT: custom DNA oligo | Rev primer that flanks the CD95 gene; used to detect CD95 gene deletion; 5'-TGGGACTCATGGGTTAAATAGAAT-3' |
| Sequence-based reagent | CD95 internal Rev primer | IDT | IDT: custom DNA oligo | Rev internal primer that targets within the CD95 gene; used to detect CD95 gene deletion; 5'-GACCAGTCTTCTCATTTCAGAGGT-3' |
| Sequence-based reagent | siScr/siNT1 | IDT; | IDT: custom DNA oligo | control non-targeting siRNA; sense: UGGUUUACAUGUCGACUAA-3' |
| Sequence-based reagent | c7/1 | IDT | custom siRNA; antisense strand corresponds to cluster 7 CD95L sequence | antisense: 5'-AUUGGGCCUG GGGAUGUUU-3'; antisense strand designed with 3' deoxy AA; complementary sense strand has 3' deoxy TT and 2'-O-methylation at the first two positions |
| Sequence-based reagent | c7/2 | IDT | custom siRNA; antisense strand corresponds to cluster 7 CD95L sequence | antisense: 5'-CCUGGGGAU GUUUCAGCUC-3'; antisense strand designed with 3' deoxy AA; complementary sense strand has 3' deoxy TT and 2'-O-methylation at the first two positions |
| Sequence-based reagent | c11 | IDT | custom siRNA; antisense strand corresponds to cluster 11 CD95L sequence | antisense: 5'-CCAACUCAAGG UCCAUGCC-3'; antisense strand designed with 3' deoxy AA; complementary sense strand has 3' deoxy TT and 2'-O-methylation at the first two positions |
| Sequence-based reagent | c15/1 | IDT | custom siRNA; antisense strand corresponds to cluster 15 CD95L sequence | antisense: 5'-AAACUGGGCUGU ACUUUGU-3'; antisense strand designed with 3' deoxy AA; complementary sense strand has 3' deoxy TT and 2'-O-methylation at the first two positions |
| Sequence-based reagent | c15/2 | IDT | custom siRNA; antisense strand corresponds to cluster 15 CD95L sequence | antisense: 5'- AACUGGGCUGU ACUUUGUA-3'; antisense strand designed with 3' deoxy AA; complementary sense strand has 3' deoxy TT and 2'-O-methylation at the first two positions |
| Sequence-based reagent | c16/1 | IDT | custom siRNA; antisense strand corresponds to cluster 16 CD95L sequence | antisense: 5'- CAACAACCUGCC CCUGAGC-3'; antisense strand designed with 3' deoxy AA; complementary sense strand has 3' deoxy TT and 2'-O-methylation at the first two positions |
| Sequence-based reagent | c16/2 | IDT | custom siRNA; antisense strand corresponds to cluster 16 CD95L sequence | antisense: 5'- AACUCUAAGCG UCCCCAGG-3'; antisense strand designed with 3' deoxy AA; complementary sense strand has 3' deoxy TT and 2'-O-methylation at the first two positions |
| Sequence-based reagent | c21 | IDT | custom siRNA; antisense strand corresponds to cluster 21 CD95L sequence | antisense: 5'- UCAACGUAUC UGAGCUCUC-3'; antisense strand designed with 3' deoxy AA; complementary sense strand has 3' deoxy TT and 2'-O-methylation at the first two positions |

*Continued on next page*

*Continued*

| Reagent type (species) or resource | Designation | Source or reference | Identifiers | Additional information |
|---|---|---|---|---|
| Sequence-based reagent | c22 | IDT | custom siRNA; antisense strand corresponds to cluster 22 CD95L sequence | antisense: 5'- AAUCUCAGACG UUUUUCGG-3'; antisense strand designed with 3' deoxy AA; complementary sense strand has 3' deoxy TT and 2'-O-methylation at the first two positions |
| Sequence-based reagent | siCtr pool | Dharmacon | D-001810–10 | control non-targeting siRNA pool |
| Sequence-based reagent | SMARTpool siRNA targeting AGO2 | Dharmacon | L-004639-00-0005 | siRNA pool designed to target AGO2 |
| Sequence based reagent (human) | GAPDH primer | Thermofisher Scientific | Hs00266705_g1 | RT-qPCR; control probe |
| Sequence based reagent (human) | CD95L primers | Thermofisher Scientific | Hs00181226_g1; Hs00181225_m1 | RT-qPCR |
| Sequence based reagent (human) | CD95 primers | Thermofisher Scientific | Hs00531110_m1; Hs00236330_m1 | RT-qPCR |
| Sequence based reagent (human) | CD95L$^{SIL}$ primer | Thermofisher Scientific | assay ID: APNKTUD | Custom RT-qPCR primer designed using the Thermofisher Scientific design tool to detect CD95L$^{SIL}$ mRNA |
| Sequence based reagent (human) | Cluster 8 CD95L small RNA primer | Thermofisher Scientific | custom probe Assay ID: CT7DPEM | Custom RT-qPCR primer designed using the Thermofisher Scientific design tool at https://www.thermofisher.com/order/custom-genomic-products/tools/small-rna to specifically detect small RNAs from cluster 8 of CD95L (5'- AAGGAGCTGGCAGAACTCCGAGA-3') |
| Sequence based reagent (human) | Cluster 21 CD95L small RNA primer | Thermofisher Scientific | custom probe Assay ID: CTAAADA | Custom RT-qPCR primer designed using the Thermofisher Scientific design tool at https://www.thermofisher.com/order/custom-genomic-products/tools/small-rna to specifically detect small RNAs from cluster 21 of CD95L (5'- TCAACGTATCTGAGCTCTCTC-3') |
| Sequence based reagent (human) | z30 primer | Thermofisher Scientific | ThermoFisher Scientific #4427975 | RT-qPCR for small RNA; control probe |
| Peptide, recombinant protein | Flag-GST-T6B peptide | PMID: 26351695 | NA | Peptide derived from GW182/TNRC6B used to pull down AGO1 to 4 |
| Commercial assay or kit | anti-Flag M2 magnetic beads | Sigma-Aldrich | Sigma-Aldrich #M8823 | |

## Reagents and antibodies

All reagents and antibodies were described previously (*Putzbach et al., 2017*) except those referenced in the following paragraphs.

## Cell lines

HeyA8 (RRID:CVCL_8878) and HeyA8 CD95 knock-out cells, HCT116 (ATCC #CCL-247; RRID:CVCL_0291) and HCT116 Drosha knock-out and Dicer knock-out cells, MCF-7 cells (ATCC #HTB-22; RRID:CVCL_0031), and 293T (ATCC #CRL-3216; RRID:CVCL_0063) cells were cultured as described previously (*Putzbach et al., 2017*). The MCF-7 CD95 knock-out and deletion cells were cultured in RPMI 1640 medium (Cellgro #10–040 CM), 10% heat-inactivated FBS (Sigma-Aldrich), 1% L-glutamine (Mediatech Inc), and 1% penicillin/streptomycin (Mediatech Inc). H460 (ATCC #HTB-177; RRID:CVCL_0459) cells were cultured in RPMI1640 medium (Cellgro Cat#10–040) supplemented with 10% FBS (Sigma Cat#14009C) and 1% L-Glutamine (Corning Cat#25–005). 3LL cells (ATCC #CRL-1642; RRID:CVCL_4358) were cultured in DMEM medium (Gibco Cat#12430054) supplemented with 10%

FBS and 1% L-Glutamine. Mouse hepatocellular carcinoma cells M565 cells were described previously (*Ceppi et al., 2014*) and cultured in DMEM/F12 (Gibco Cat#11330) supplemented with 10% FBS, 1% L-Glutamine and ITS (Corning #25–800-CR). All cell lines were authenticated using STR profiling and tested monthly for mycoplasm using PlasmoTest (Invitrogen).

## Plasmids and constructs

The pLenti-CD95L was synthesized by sub-cloning an insert containing the CD95L ORF (NM_000639.2; synthesized by IDT as minigene with flanking 5' NheI RE site and 3' XhoI RE sites in pIDTblue vector) into the pLenti-GIII-CMV-RFP-2A-Puro vector (ABM Inc). The insert and the backbone were digested with NheI (NEB #R0131) and XhoI (NEB #R0146) restriction enzymes. Subsequent ligation with T4 DNA ligase created the pLenti-CD95L vector. The pLenti-CD95L$^{MUT}$ vector was created by sub-cloning a CD95L cDNA insert with two nucleotide substitutions in codon 218 (*TAT - > CGT)* resulting in replacement of tyrosine for arginine, which has been described to inhibit binding to CD95 (*Schneider et al., 1997*) into the pLenti-GIII-CMV-RFP-2A-Puro vector. The pLenti-CD95L$^{MUT}$NP vector was created by inserting a CD95L ORF cDNA sequence containing both the Y218R mutation and a single nucleotide substitution at the second codon (*CAG - > TAG*), resulting in a premature stop codon right after the start codon, into the pLenti-GIII-CMV-RFP-2A-Puro vector. The pLenti-CD95L$^{SIL}$ was created by sub-cloning a mutant CD95L ORF cDNA sequence with codons synonymously mutated (*Figure 1—figure supplement 2A*) to the next most highly utilized codon in human cells (exceptions were made within the proline rich domain (*Novoradovsky et al., 2005*) to meet gene synthesis design criteria) into the pLenti-GIII-CMV-RFP-2A-Puro vector. The GenScript Codon Usage Frequency Table Tool for Expression Host Organism: Human (https://www.genscript.com/tools/codon-frequency-table) was used to determine codon use frequencies.

The pLenti-CD95L$^{MUT}$NP (G258A) mutant was generated by site-directed mutagenesis. Identification of the ATG giving rise to the 24 kDa CD95L$^{MUT}$NP peptide was predicted using the Sequence Manipulation Suite (*Stothard, 2000*). The next methionine downstream of the premature stop codon was predicted to produce a 22.27 kDa protein, and thus was mutated to an isoleucine (ATG >ATA). The Agilent QuikChange Primer Design tool was used to design mutagenesis primers (*Novoradovsky et al., 2005*). The following primers were synthesized by IDT with 5' phosphorylation and were PAGE purified 5'- GCCTGTGTCTCCTTGTGATATTTTTCATGGTTCTGGTTG-3' and 5'-CAACCAGAACCATGAAAAATATCACAAGGAGACACAGGC-3'. The PCR reaction was performed using NEB Phusion High-Fidelity DNA Polymerase (#M0530) with HF Buffer (#B0518) with 3% DMSO according to manufacturer's protocol. The BioRad T100 Thermocycler was utilized for the mutagenesis reaction using the following cycling conditions: (step 1) 98°C, 30 s; (step two) 25 cycles: 98°C, 10 s; 68°C, 30 s; 72°C, 6 min; (step three) 72°C, 10 min. The template DNA was digested with DpnI (Thermo Scientific F541), and the mutagenized plasmid was ligated using T4 ligase (Thermo Scientific F541) and transformed into Stbl3 (Invitrogen C737303). Successful mutants were screened by growth on Kanamycin plates, and confirmed by Sanger sequencing (ACGT, Inc).

The pLenti-CD95L 5' fragment vector was created by subcloning the 5' fragment of the CD95L$^{MUT}$NP ORF cDNA sequence (5'-GCTAGCCTGACTCACCAGCTGCCATGTAGCAGCCCTTCAATTACCCATATCCCCAGATCTA CTGGGTGGACAGCAGTGCCAGCTCTCCCTGGGCCCCTCCAGGCACAGTTCTTCCCTGTCCAACCTCTGTGCCCAGAAGGCCTGGTCAAAGGAGGCCAC-CACCACCACCGCCACCGCCACCACTACCACCTCCGCCGCCGCCGCCACCACTGCCTCCACTACCGCTGCCACCCCTGAAGAAGAGAGGGAACCACAGCACAGGCCTGTGTCTCCTTGCTCGAG-3') into the pLenti-GIII-CMV-RFP-2A-Puro vector. The pLenti-CD95L 3' fragment vector was created by subcloning the 3' half of the CD95L$^{MUT}$NP ORF cDNA sequence (5'-GCTAGCCCTGGGGATGTTTCAGCTCTTCCACCTACAGAAGGAGCTGGCAGAACTCCGAGAGTCTACCAGCCAGATGCACA-CAGCATCATCTTTGGAGAAGCAAATAGGCCACCCCAGTCCACCCCCTGAAAAAAAGGAGCTGAGGAAAGTGGCCCATTTAACAGGCAAGTCCAACTCAAGGTCCATGCCTCTGGAATGGGAAGA-CACCTATGGAATTGTCCTGCTTTCTGGAGTGAAGTATAAGAAGGGTGGCCTTGTGATCAATGAAACTGGGCTGTACTTTGTATATTCCAAAGTATACTTCCGGGGTCAATCTTGCAACAACCTGCCCCTGAGCCACAAGGTCTACATGAGGAACTCTAAGCGTCCCCAGGATCTGGTGATGATGGAGGGGAAGATGATGAGCTACTGCACTACTGGGCAGATGTGGGCCCGCAGCAGCTACCTGGGGGCAGTGTTCAATCTTACCAGTGCTGATCATTTATATGTCAACGTATCTGAGCTCTCTCTGGTCAATTTTGAGGAATCTCAGACGTTTTTCGGCTTATATAAGCTCTAAGAGAAGCACTTTGGGATCTCGAG-3') into the pLenti-GIII-CMV-RFP-2A-Puro vector.

## Overexpression of CD95L cDNAs

All lentiviral constructs were generated in 293T cells as described previously (*Putzbach et al., 2017*). HeyA8 and MCF-7 (and all derivative cell lines) cells overexpressing wild type CD95L and mutant CD95L cDNAs were generated by seeding cells at 100,000 cells per well in a 6-well plate and infecting cells with lentivirus generated in 293T cells (500 µl viral supernatant per well) with 8 µg/ml Polybrene and subsequent centrifugation at 2700 rpm for 0.5–1 hr. Media was changed next day, and confluent wells were expanded to larger dishes. Selection was started on following day with 3 µg/ml puromycin. HCT116, HCT116 Drosha knockout, and HCT116 Dicer knockout cells (*Kim et al., 2016*) overexpressing CD95L cDNAs were generated by seeding cells at 100,000 cells per well in a 24-well plate or 500,000 cells per well in a 6-well plate and infecting cells with lentivirus generated in 293T cells (100 µl virus per 24-well or 500 µl per 6-well) in the presence of 8 µg/ml Polybrene. Media was changed the next day, and cells were selected with 3 µg/ml puromycin the following day. Infection with empty pLenti was always included as a control.

To assess toxicity of overexpressing CD95L cDNAs, cells infected with these constructs were plated in a 96-well plate 1 day after selection in the presence of puromycin (uninfected cells were all dead after 1 day in presence of puromycin); Cell confluency was assessed over time using the Incu-Cyte as described previously (*Putzbach et al., 2017*).

To assess overexpression of CD95L cDNAs in apoptosis-sensitive HeyA8 cells in *Figure 1A*, infections with CD95L lentiviruses were done in 96-well plate using 50 µl of virus in the presence of 20 µM zVAD-fmk (Sigma-Aldrich #V116) and 8 µg/ml Polybrene; media was changed next day in the presence of 20 µM zVAD-fmk; 3 µg/ml puromycin was added the following day. Infection with the CD95L constructs for the RT-qPCR and Western blot in *Figure 1B* were done in a 6-well plate in the presence of 20 µM zVAD-fmk.

For the experiment in *Figure 3B*, *Figure 3—figure supplement 1*, and *Figure 5G*, cells were reverse transfected in a 6-well plate; 100,000 cells were plated in wells with either the On-TargetPlus non-targeting siRNA (Dharmacon #D-001810–10) or siAGO2 pool (Dharmacon ##L-004639-00-0005) at 25 nM complexed with 1 µl RNAiMax. After ~24 hr, the cells were infected with either pLenti or different pLenti-CD95L WT or mutant constructs (500 µl of viral supernatant [25% of total volume] as described above. Next day, media was replaced, and cells were expanded to 10 cm plates. The following day, 3 µg/ml puromycin was added. When puromycin selection was complete (one day later), 750–3000 cells were plated per well in a 96-well plate and put in the IncuCyte machine to assess cell confluency over time. When wt HeyA8 cells were infected cells were plated one hour after infection and centrifugation. Puromycin was added to the wells 50 hr after infection.

## CRISPR deletions

We co-transfected a Cas9-expressing plasmid (*Jinek et al., 2013*) and two gRNAs that target upstream and downstream to delete an entire section of DNA as described previously (*Putzbach et al., 2017*). The gRNA scaffold was used as described (*Mali et al., 2013*). The gRNAs were designed using the algorithm found at http://crispr.mit.edu; only gRNAs with a score above 50 were considered.

A deletion of 227 nucleotides in exon 4 of CD95 in MCF-7 cells (ΔshR6, clone #21) was generated using gRNAs described previously (*Putzbach et al., 2017*). Deletion of this site results in a frameshift mutation that causes a protein-level knock-out (*Putzbach et al., 2017*). PCR with flanking external primers (Fr: 5'-*GGTGTCATGCTGTGACTGTTG*-3' and Rev: 5'-*TTTAGCTTAAGTGGCCAGCAA*-3') and internal primers (Fr primer and the internal Rev primer 5'-*AAGTTGGTTTACATCTGCAC*-3') was used to screen for single cell clones that harbor a homozygous deletion.

The two sequences targeted by the flanking gRNAs for the deletion of the entire CD95 gene were 5'-GTCAGGGTTCGTTGCACAAA-3' and 5'-TGCTTCTTGGATCCCTTAGA-3'. For detection of the CD95 gene deletion, the flanking external primers were 5'-*TGTTTAATATAGCTGGGGCTATGC*-3' (Fr primer) and 5'-*TGGGACTCATGGGTTAAATAGAAT*-3' (Rev primer), and the internal reverse primer was 5'-*GACCAGTCTTCTCATTTCAGAGGT*-3'. After screening the clones, Sanger sequencing was performed to confirm the proper deletion had occurred.

## Real-Time quantitative PCR

The relative expression of specific mRNAs was quantified as described previously (*Gao et al., 2018*). The primer/probes purchased from ThermoFisher Scientific were GAPDH (Hs00266705_g1), human CD95L (Hs00181226_g1 and Hs00181225_m1), human CD95 (Hs00531110_m1 and Hs00236330_m1), and a custom primer/probe to detect the CD95L$^{SIL}$ mRNA (designed using the Thermofisher Scientific custom design tool; assay ID: APNKTUD) and CD95L 5' fragment mRNA (designed using the Thermofisher Scientific custom design tool; assay ID: APEPTMU).

Custom RT-qPCR probes designed to specifically detect sRNA species were used to detect CD95L fragments in *Figure 4G*. Abundant Ago bound CD95L sRNA sequences were identified from Drosha k.o. Ago pull-down sequencing data. sRNA sequences mapping to *FASLG* were sorted by their 3' end position. The number of sRNA reads terminating at the same nucleotide position were counted, and the most abundant sequences were selected for qPCR probe design from clusters 8 and 21. These probes were designed using ThermoFisher's Custom TaqMan Small RNA Assay Design Tool (https://www.thermofisher.com/order/custom-genomic-products/tools/small-rna/) to target the cluster eight sequence (5'- AAGGAGCTGGCAGAACTCCGAGA-3') and the cluster 21 sequence (5'- TCAACGTATCTGAGCTCTCTC-3'). Detection of these fragments involves a two-step amplification protocol used to detect miRNAs. In the first step, the High-Capacity cDNA reverse transcription kit is used to selectively reverse transcribe the two clusters to be quantified using specific primers and 100 ng RNA input. The cDNA was diluted 1:5. The qPCR reaction mixture is composed of the diluted cDNA, the custom probes, and the Taqman Universal PCR Master Mix (Applied Biosystems #43240018). Reactions were performed in triplicate. Ct values were determined using the Applied Biosystems 7500 Real Time PCR system with a thermocycle profile of 50°C for two min (step one), 95°C for 10 min (step two), and then 40 cycles of 95°C for 15 s (step three) and 60°C for 1 min (step four). The ΔΔCt values between the sRNA of interest and the control were calculated to determine relative abundance of the sRNA. Samples were normalized to Z30 (ThermoFisher Scientific #4427975).

## Western blot analysis

Detection of human CD95, CD95L, AGO1 and AGO2 was done via Western blot as described previously (*Putzbach et al., 2017*).

## CD95 surface staining

Flow cytometry was used to quantify the level of membrane-localized CD95 as described previously (*Putzbach et al., 2017*).

## Cell death quantification (DNA fragmentation) and ROS production

The percent of subG1 nuclei (fragmented DNA) was determined by PI staining/flow cytometry as described previously (*Putzbach et al., 2017*). ROS production was quantified using the cell-permeable indicator 2',7'-dichlorodihydrofluorescein diacetate (ThermoFisher Scientific #D399) as previously described (*Hadji et al., 2014*).

## Assessing cell growth and fluorescence over time

After treatment/infection, 750–3000 cells were seeded in a 96-well plate at least in triplicate. Images were captured at indicated time points using an IncuCyte ZOOM live cell imaging system (Essen BioScience) with a 10x objective lens. Percent confluence and total fluorescent integrated intensity was calculated using the IncuCyte ZOOM software (version 2015A).

## Infection of cells for Ago-pull down and small RNA-Seq analysis

HeyA8 ΔshR6 clone #11 cells were seeded at 75,000 cells per well on 6-well plates, and the HCT116 and HCT116 Drosha knock-out cells were both seeded at 500,000 per well on 6-well plates. The HeyA8 ΔshR6 clone #11 cells were infected with 0.5 mL of empty pLenti or pLenti-CD95L-WT viral supernatant per well. The HCT116 and HCT116 Drosha knockout cells were infected with 0.5 mL empty pLenti or pLenti-CD95L$^{MUT}$NP viral supernatant per well. Media was changed the next day and the cells were pooled and expanded to multiple 15 cm dishes. Selection with 3 µg/mL puromycin began the following day. The next day, the HeyA8 ΔshR6 clone #11 infected cells were seeded

at 600,000 cells per dish in multiple 15 cm dishes; the HCT116 and HCT116 Drosha knock-out cells were seeded at 5 million cells per dish in multiple 15 cm dishes. Two days later, each of the samples were pelleted and split in two: one pellet was lysed and processed for small RNA sequencing, and the other pellet was flash frozen in liquid nitrogen. The pellets were stored at −80°C until they could be used for the Ago pull-down experiment. The purpose of splitting the sample was so that we could compare the total cellular pool of sRNAs to the fraction that was bound to the RISC. This way, the processing CD95L-derived fragments from the full-length mRNA in the cytosol to the final mature RISC-bound form could be mapped. This was all done in duplicate.

## RNA-Seq analysis

Total RNA was isolated using the miRNeasy Mini Kit (Qiagen, #74004) following the manufacturer's instructions. An on-column digestion step using the RNase-free DNase Set (Qiagen #79254) was included. Both small and large mRNA libraries were generated and sequenced as described previously (*Putzbach et al., 2017*). Reads were trimmed with TrimGalore and then aligned to the hg38 assembly of the human genome with Tophat. Raw read counts were assigned to genes using HTSeq and differential gene expression was analyzed with the R Bioconductor EdgeR package (*Robinson et al., 2010*).

## Ago pull down and RNA-Seq analysis of bound sRNAs

Cell pellets were harvest at 50 hr after plating (122 hr after infection) and were flash frozen in liquid nitrogen. The pellets were stored at −80°C until ready for further processing. Between 10 and 25 × $10^6$ cells were lysed in NP40 lysis buffer (20 mM Tris, pH 7.5, 150 mM NaCl, 2 mM EDTA, 1% (v/v) NP40, supplemented with phosphatase inhibitors) on ice for 15 min. The lysate was sonicated 3 times for 30 s at 60% amplitude (Sonics, VCX130) and cleared by centrifugation at 12,000 g for 20 min. AGO1-4 were pulled down by using 500 μg of Flag-GST-T6B peptide (*Hauptmann et al., 2015*) and with 60 μl anti-Flag M2 magnetic beads (Sigma-Aldrich) for 2 hr at 4°C. The pull-down was washed three times in NP40 lysis buffer. During the last wash, 10% of beads were removed and incubated at 95°C for 5 min in 2x SDS-PAGE sample buffer. Samples were run on a 4–12% SDS-PAGE and transferred to nitrocellulose membrane. The pull-down efficiency was determined by immunoblotting against AGO1 (Cell Signaling #5053; RRID:AB_10695871 and Abcam #98056; RRID: AB_10680548) and AGO2 (Abcam #32381; RRID:AB_867543). To the remaining beads 500 μl TRIzol reagent were added and the RNA extracted according to the manufacturer's instructions. The RNA pellet was diluted in 20 μl of water. The sample was split, and half of the sample was dephosphory-lated with 0.5 U/μl of CIP alkaline phosphatase at 37°C for 15 min and subsequently radiolabeled with 0.5 μCi γ-$^{32}$P-ATP and 1 U/μl of T4 PNK kinase for 20 min at 37°C. The AGO1-4 interacting RNAs were visualized on a 15% urea-PAGE. The remaining RNA was taken through a small RNA library preparation as previously described (*Hafner et al., 2012*). Briefly, RNA was ligated with 3' adenylated adapters and separated on a 15% denaturing urea-PAGE. The RNA corresponding to insert size of 19–35 nt was eluted from the gel, ethanol precipitated followed by 5' adapter ligation. The samples were separated on a 12% Urea-PAGE and extracted from the gel. Reverse transcription was performed using Superscript III reverse transcriptase and the cDNA amplified by PCR. The cDNA was sequenced on Illumina HiSeq 3000. Adapter sequences: Adapter 1 – NNTGACTGTGGAA TTCTCGGGTGCCAAGG; Adapter 2 – NNACACTCTGGAATTCTCGGGTGCCAAGG, Adapter 3 – NNACAGAGTGGAATTCTCGGGTGCCAAGG, Adapter 4 – NNGCGATATGGAATTCTCGGG TGCCAAGG, Adapter 47 – NNTCTGTGTGGAATTCTCGGGTGCCAAGG, Adapter 48 – NNCAGCA TTGGAATTCTCGGGTGCCAAGG, Adapter 49 – NNATAGTATGGAATTCTCGGGTGCCAAGG, Adapter 50 – NNTCATAGTGGAATTCTCGGGTGCCAAGG. RT primer sequence: GCC TTGGCACCCGAGAATTCCA; PCR primer sequences: CAAGCAGAAGACGGCATACGAGATCGTGA TGTGACTGGAGTTCCTTGGCACCCGAGAATTCCA. To identify CD95L-derived sRNAs among the sequenced reads, a BLAST database was generated from each set of reads, and blastn was used to query the CD95L ORF (derived from NM_000639.2) against reads from cells infected with pLenti-CD95L and to query the CD95L$^{MUT}$NP ORF sequence against reads from cells infected with CD95L$^{MUT}$NP. The only reads considered further were those matching a CD95L sequence with an e-value of less than 0.05% and 100% identity across the entire length of the read. This resulted in the loss of a few reads less than 19/20 nt in length. The filtered BLAST hits were converted to a bed

formatted file, describing the locations of reads relative to the relevant CD95L sequence, and the R package Sushi was used to plot the bed files and generate *Figure 4B–E*.

## Assessing toxicity of CD95L-derived sRNAs

To determine whether guide RNAs derived from the over-expressed CD95L mRNA could evoke toxicity, the small CD95L-derived RNA reads (corresponding to different clusters shown in *Figure 4C*) bound to AGO from the HCT116 Drosha knock-out cells were converted to siRNAs. First, all reads less than 18 nucleotides were filtered out, as these do not efficiently incorporate into the RISC. siRNAs were designed with antisense strands fully complementary to the CD95L-derived sequences that mapped to areas of the CD95L mRNA secondary structure (*Figure 4—figure supplement 1A*) that are predicted to form duplexes. These sequences were designed as 19 nucleotide oligos with a 3′ deoxy AA. The complementary sense strand was designed with a 3′ deoxy TT and 2′-O-methylation at the first two positions to prevent its incorporation into the RISC. These oligos were ordered from IDT and annealed to form the final siRNAs. The sequences of the antisense strands (corresponding to the CD95L mRNA-derived cluster fragments) were as follows: 5′-AUUGGGCCUGGGGA UGUUU-3′ (c7/1), 5′-CCUGGGGAUGUUUCAGCUC-3′ (c7/2), 5′-CCAACUCAAGGUCCAUGCC-3′ (c11), 5′-AAACUGGGCUGUACUUUGU-3′ (c15/1), 5′- AACUGGGCUGUACUUUGUA-3′ (c15/2), 5′-CAACAACCUGCCCCUGAGC-3′ (c16/1), 5′- AACUCUAAGCGUCCCCAGG-3′ (c16/2), 5′- UCAACG UAUCUGAGCUCUC-3′ (c21), and 5′- AAUCUCAGACGUUUUUCGG-3′ (c22).

These eight siRNAs were reverse transfected into HeyA8, H460, M565, and 3LL cells using RNAi-MAX transfection reagent (ThermoFisher Scientific) at 10 nM in triplicate as previously described (*Murmann et al., 2018a*). The non-targeting (NT) and siL3 siRNAs, as described previously (*Putzbach et al., 2017*), were used as a negative and positive control, respectively. Cell death was quantified via ATP release 96 hr after transfection using CellTiter-Glo (Promega). The % viability was calculated in relation to the RNAiMAX-only treatment structure (*Figure 4—figure supplement 1B*).

To analyze the distribution of seed toxicity of all the sRNAs bound to RISC in wt and Drosha k.o. cells, read-based Ago pull down data were used. 6mer seed sequences (position 2–7) were extracted from all unique sequences present in wt or Drosha k.o. cells. The corresponding average 6mer seed toxicity (between HeyA8 and H460 cells) was added for each unique sequence. These sequences were aggregated into six groups according to their different levels of seed toxicity (numbers shown are percentage cell viability): (1)<20%; (2) 20 ~ 40%; (3) 40 ~ 60%; (4) 60 ~ 80%; (5) 80 ~ 100%; (6)>100%. Total reads (RPM) in each seed toxicity group were added up and plotted (*Figure 5F*). In addition, the distribution of seed toxicity of all miRNAs bound to the RISC in wt and Drosha k.o. cells were analyzed in a similar manner except that only reads that could be aligned to miRNAs were included.

## Analysis of the CD95L^SIL mutant's theoretical toxicity

All possible 6mers that could be generated through processing of the 846 nt sequence of either CD95L WT or CD95L^SIL were extracted in R, a total of 841 possible 6mers. The corresponding HeyA8 viability data from the 4096 6mer seed screen was joined to the corresponding theoretical CD95L 6mers (*Gao et al., 2018*). A density plot was generated from the viability data. R packages used: dplyr, data.table, ggplot2. A Two-sample Kolmogorov-Smirnov test was performed in R to determine statistical significance.

## Analysis of RISC bound sRNAs derived from coding genes

All reads from Ago pull downs of Drosha k.o. samples with either pLenti or CD95L over-expressed were aligned to the hg38 genome as previously described (*Putzbach et al., 2017*) (see *Figure 5— figure supplement 2*). Reads were extracted that aligned uniquely to the genome for each pair of replicates, and pooled into one set per sample type (CD95L or pLenti). Identical reads were collapsed and counted and reads <15 bp or >50 bp in length were eliminated, as well as reads containing N's. The number of reads found at least 10 times across each pair of replicate samples were 10,839 for CD95L expressing cells and 10,617 for pLenti infected cells. The results were filtered so that only 100% matches across the entire length of the read were counted, in the direction of transcription. The start position was reported for each uniquely mapping, highly abundant (>=10 copies) read. The number of such unique start positions was counted to obtain a putative number of stacks

for each transcript. Any gene with at least one stack was counted, and the ten genes with the highest number of stacks are plotted in *Figure 5B* (and *Figure 5—figure supplement 1B*). Once the ten genes that contained the most RISC bound reads in the CD95L expressing cells were identified, a single transcript was pulled out for each gene. In each case the transcript with the longest 3'UTR was used: 1 - ZC3HAV1 (ENST00000242351), 2 - FAT1 (ENST00000441802), 3 - SRRM2 (ENST00000301740), 4 - TNFRSF10D (ENST00000312584), 5 - PABPC1 (ENST00000318607), 6 - CCND1 (ENST00000227507), 7 - MYC (ENST00000621592), 8 - YWHAG (ENST00000307630), 9 - DKK1 (ENST00000373970), 10 - AKAP12 (ENST00000402676). These reads were blasted against all of the raw RNA-seq data. Only 100% matches across the entire length of the read were considered, in the direction of transcription, but all reads that matched the transcript at that stringency were plotted, regardless of their individual abundance or uniqueness, with each individual blue line representing an individual read, with its length in the plot proportional to the read length.

## Determine the abundance of the coding genes that give rise to Ago-bound reads

To determine the abundance of coding genes with Ago bound reads (*Figure 5C*), we isolated all genes that had an RPKM of at least 10 using uniquely mapped reads from a conventional long RNA-Seq data set. These resulted in 4262 genes in the CD95L expressing HCT116 Drosha k.o. cells and 4256 genes in the pLenti infected cells. Both data sets were very similar with >90% overlap of the two sets.

## Statistical analyses

Continuous data were summarized as means and standard deviations (except for all IncuCyte experiments where standard errors are shown) and dichotomous data as proportions. Continuous data were compared using t-tests for two independent groups and one-way ANOVA for three or more groups. For evaluation of continuous outcomes over time, two-way ANOVA was used with one factor for the treatment conditions of primary interest and a second factor for time treated as a categorical variable to allow for non-linearity.

The effects of treatment on wild-type versus Drosha knock-out cells were statistically assessed by fitting regression models that included linear and quadratic terms for value over time, main effects for treatment and cell type, and two- and three-way interactions for treatment, cell-type and time. The three-way interaction on the polynomial terms with treatment and cell type was evaluated for statistical significance since this represents the difference in treatment effects over the course of the experiment for the varying cell types.

GSEA used in *Figure 2B* was performed using the GSEA v2.2.4 software from the Broad Institute (http://software.broadinstitute.org/gsea); 1000 permutations were used. The Sabatini gene lists were set as custom gene sets to determine enrichment of survival genes versus the nonsurvival control genes in downregulated genes from the RNA-Seq data as done previously (*Putzbach et al., 2017*); p-values below 0.05 were considered significantly enriched. Genes with an average normalized read expression (across both pair of duplicates) below three were excluded so as to only include genes that are truly expressed. The GO enrichment analysis shown in *Figure 2D* was performed with all genes that after alignment and normalization were found to be at least 1.5 fold downregulated with an adjusted p-value of <0.05 using the software available on www.Metascape.org and default running parameters. The other data sets used in this analysis (HeyA8 cells transfected with a toxic siRNA targeting CD95L siL3 and 293T infected with toxic shRNAs targeting CD95L shL1 and shL3 and HeyA8 cells infected with a toxic shRNA targeting CD95 shR6) were previously described (*Putzbach et al., 2017*).

All statistical analyses were conducted in Stata 14 or R 3.3.1.

### Data availability

RNA sequencing data generated for this study is available in the GEO repository: GSE103631 (https://www.ncbi.nlm.nih.gov/geo/query/acc.cgi?acc=GSE103631) and GSE114425 (https://www.ncbi.nlm.nih.gov/geo/query/acc.cgi?acc=GSE114425).

## Acknowledgements

We are grateful to Siquan Chen for testing small CD95L-derived sRNAs. MH and AAS were supported by the Intramural Research Program of NIAMS. AAS acknowledges support by the Swedish

Research Council postdoctoral fellowship. This work was funded by training grant T32CA009560 (to WP and AHK), R50CA221848 (to ETB), and R35CA197450 (to MEP).

## Additional information

### Funding

| Funder | Grant reference number | Author |
| --- | --- | --- |
| National Institutes of Health | R35CA197450 | Marcus E Peter |
| National Institutes of Health | T32CA009560 | Will Putzbach Ashley Haluck-Kangas |
| National Institutes of Health | R50CA221848 | Elizabeth T Bartom |

The funders had no role in study design, data collection and interpretation, or the decision to submit the work for publication.

### Author contributions

Will Putzbach, Conceptualization, Formal analysis, Investigation, Methodology, Writing—review and editing; Ashley Haluck-Kangas, Formal analysis, Investigation, Writing—review and editing; Quan Q Gao, Aishe A Sarshad, Austin Stults, Abdul S Qadir, Formal analysis; Elizabeth T Bartom, Data curation, Formal analysis; Markus Hafner, Supervision; Marcus E Peter, Conceptualization, Formal analysis, Supervision, Funding acquisition, Investigation, Writing—original draft, Project administration, Writing—review and editing

### Author ORCIDs

Will Putzbach http://orcid.org/0000-0002-2669-6654
Ashley Haluck-Kangas http://orcid.org/0000-0002-9861-6801
Quan Q Gao http://orcid.org/0000-0002-3316-2968
Elizabeth T Bartom http://orcid.org/0000-0002-5618-2582
Markus Hafner http://orcid.org/0000-0002-4336-6518
Marcus E Peter http://orcid.org/0000-0003-3216-036X

### Decision letter and Author response

Decision letter https://doi.org/10.7554/eLife.38621.028
Author response https://doi.org/10.7554/eLife.38621.029

## Additional files

### Supplementary files

• Supplementary file 1. Reads from coding and noncoding genes pulled-down with Ago proteins in HCT116 Drosha k.o. pLenti-CD95L cells. Tab 1: Reads from all genes; Tab 2: Reads from processed protein coding genes (>10 reads), Tab 3: Reads from unprocessed protein coding genes (>10 reads).
DOI: https://doi.org/10.7554/eLife.38621.020

• Transparent reporting form
DOI: https://doi.org/10.7554/eLife.38621.021

### Data availability

Sequencing data have been deposited in GEO under accession codes: GSE103631 and GSE114425.

The following datasets were generated:

| Author(s) | Year | Dataset title | Dataset URL | Database, license, and accessibility information |
| --- | --- | --- | --- | --- |
| Putzbach W, Peter | 2018 | CD95/Fas ligand mRNA is toxic to | https://www.ncbi.nlm. | Publicly available at |

| ME, Bartom E | | cells | | nih.gov/geo/query/acc. cgi?acc=GSE103631 | the NCBI Gene Expression Omnibus (accession no. GSE10 3631) |
| Putzbach WE, Ha-luck-Kangas A, Gao QQ, Sarshad AA, Bartom E, Stults A, Qadir AS, Scholtens DM, Hafner M, Peter ME | 2018 | CD95L mRNA is toxic to cells | | https://www.ncbi.nlm. nih.gov/geo/query/acc. cgi?acc=GSE114425 | Publicly available at the NCBI Gene Expression Omnibus (accession no. GSE114425) |

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
