## [Decision Letter]

Thank you for submitting your article "CD95L mRNA is toxic to cells" for consideration by *eLife*. Your article has been reviewed by two peer reviewers, and the evaluation has been overseen by Jeffrey Settleman as the Senior Editor and Reviewing Editor. The reviewers have opted to remain anonymous.

The reviewers have discussed the reviews with one another and the Reviewing Editor has drafted this decision to help you prepare a revised submission.

Summary:

The manuscript extends this group's previous observations that a large number of shRNA and siRNA sequences generated against CD95 and CD95L promote cellular toxicity. This new study reports that transfection of CD95L mRNA is also sufficient to kill cells, independently of CD95L protein production, involving the generation of small RNA species that are loaded into a Ago2/RISC complex. Overall, the manuscript is well-written, the data are presented clearly, and experimental approaches are sound. Though the authors have painted the picture of a unique, biologically-relevant phenomenon, key missing data points must be addressed while other details should be clarified in order to provide full support for their conclusions. Moreover, while the findings are provocative and potentially interesting, they are somewhat incremental relative to the previous study. Several additional experiments would need to be completed to make this study relevant to the readers of *eLife*.

Essential revisions:

1) The authors should repeat the experiment in Figure 3B with the mutant RNA forms (SIL and NP) to prove Ago2/RISC dependence. I would also suggest the authors perform this experiment in CD95 wt cells, to show that Ago2 KD only rescues viability in the absence of CD95, while in wt the CD95/CD95L interaction remains intact and toxic (the mutant RNAs should perform somewhat similarly in wt and CD95 KOs).

2) As the authors speculate in the Discussion, it would be important to know which additional mRNA species also show this property when transfected into cells? Why was only CD95L and not CD95 tested here? Simply introducing the stop codon after the first exon would be an easy way to test this hypothesis in a relatively large number of genes.

3) Similar to the above point, the authors could interrogate their existing data on the presence of additional small RNAs in the total RNA fraction used in Figure 4C to ask how many additional endogenous genes end up being processed. They could also interrogate the Ago pulldown data to see if they get incorporated into the Drosha KO RISC. Is there something special about CD95L mRNA processing, or does RISC load up with spurious RNA fragments in Drosha KOs? This is not clear from the analysis as presented. They already document the presence of VTRNA2-1, VTRA1-2 and TFCP2L1 sequences in the RISC – is there any reason to think these sequences might be toxic, or have additional cellular consequences?

4) In Figure 1—figure supplement 2, I was surprised to see that the "SIL" mutant was as active as the wt CD95L mRNAs, despite expressing >85% less protein and having ~37% (somewhat evenly distributed) nucleotide divergence. Is this due to expression of specific sRNAs that were not altered by codon editing (maybe answerable by sRNA sequencing)? Is the ~12% of protein providing this activity (maybe answerable by removing the start codon)? Subjecting this mutant construct to the secondary structure prediction could be informative. The authors should reconcile this counterintuitive result.

5) The authors identify >20 potential segments within the CD95L mRNA that correspond to sRNAs sequenced after AGO IP. Similar to the experiments shown in Figure 1A, the authors should express (in HeyA8 cells) fragments of the CD95L mRNA (100-200 nts?, overlapping with one-another) in order to identify regions with the most toxic profile. If sensitization is necessary to see a phenotype, this experiment could be performed in the HCT116 wt and Drosha KO models. This is critical to understand if specific sequences (derived from the mRNA, not expressed separately via shRNA) direct the phenotype, and if local or broad RNA structure impacts sRNA biogenesis.

---

## [Author Response]

Summary:The manuscript extends this group's previous observations that a large number of shRNA and siRNA sequences generated against CD95 and CD95L promote cellular toxicity. This new study reports that transfection of CD95L mRNA is also sufficient to kill cells, independently of CD95L protein production, involving the generation of small RNA species that are loaded into a Ago2/RISC complex. Overall, the manuscript is well-written, the data are presented clearly, and experimental approaches are sound. Though the authors have painted the picture of a unique, biologically-relevant phenomenon, key missing data points must be addressed while other details should be clarified in order to provide full support for their conclusions. Moreover, while the findings are provocative and potentially interesting, they are somewhat incremental relative to the previous study. Several additional experiments would need to be completed to make this study relevant to the readers of eLife.

We have addressed all concerns raised by the reviewers and feel that the story now provides new insights into a potential new mechanism of gene regulation. While the research advance is meant to be an extension of work already published, our new analysis demonstrates with little doubt that not only the CD95L protein can kill cancer cells (through induction of apoptosis) but that the mRNA itself is toxic through induction of DISE. We are not aware that anybody has ever shown such an activity for any gene. In addition, our discovery that under low miRNA conditions about 3% of the protein coding human genome (mostly genes regulating cell proliferation) is processed in a way similar to CD95L mRNA and loaded into the RISC is highly novel. This finding could have relevance for situations in which miRNA expression is low such as is seen in stem cells or in advanced cancers. We noticed that the reviewers/editor referred to the CD95L-derived RNAs as sRNAs rather than siRNAs. We think that is a good idea and have adopted this terminology in the revised manuscript.

Essential revisions:1) The authors should repeat the experiment in Figure 3B with the mutant RNA forms (SIL and NP) to prove Ago2/RISC dependence. I would also suggest the authors perform this experiment in CD95 wt cells, to show that Ago2 KD only rescues viability in the absence of CD95, while in wt the CD95/CD95L interaction remains intact and toxic (the mutant RNAs should perform somewhat similarly in wt and CD95 KOs).

We have performed all these experiments as suggested. Both the NP and the SIL mutant's toxicity requires Ago2 when expressed in CD95 k.o. cells. In wt cells the SIL mutant kills rapidly through apoptosis, and is not attenuated by Ago2 kd. In contrast, the NP mutant also kills wt cells but this is dependent on Ago2. All these experiments have now been added as new Figure 3—figure supplement 1.

2) As the authors speculate in the Discussion, it would be important to know which additional mRNA species also show this property when transfected into cells? Why was only CD95L and not CD95 tested here? Simply introducing the stop codon after the first exon would be an easy way to test this hypothesis in a relatively large number of genes.

This experiment is more complicated than it appears. For instance, we recently reported on a set of genes other than CD95/CD95L that potentially contain toxic sequences (Patel and Peter, 2017). However, just taking these genes (some of which are functionally not well characterized) and adding a stop codon may result in data that are difficult to interpret. Alternative start codons downstream of the main start codon may be used to produce truncated protein and the function and interaction partners of some of these genes or their truncated peptides could be multiple. Hence, results may be difficult to interpret. In contrast, CD95L has been very well characterized by many groups including the group of the corresponding author. In fact, the main reason we discovered this new activity of the mRNA was the established fact that there is no interaction partner in a cell other than CD95 (leaving aside decoy receptor DcR3, a secreted protein that will likely not be relevant in our cells), and the well described apoptosis-inducing activity of CD95L. The combination of the canonical apoptosis induction as a robust control form of protein induced cell death and the ability of eliminating this activity completely from any cell by deleting CD95 presents a unique situation we will not easily find with any other gene.

We previously showed that the most toxic sequences are located in the ORF of CD95L and the 3'UTR of CD95 (not the ORF of CD95). This is why CD95L was chosen over CD95. Also, si/shRNAs derived from CD95L were consistently more toxic than the ones derived from CD95.

Finally, we recently determined that the form of siRNA induced toxicity we described is in fact solely dependent on the 6mer seed sequence of the siRNA loaded into the RISC. This work is in press at Nature Communications, and has also been deposited at BioRxive (Gao et al., 2018). Based on this recent analysis we have called this activity 6mer seed toxicity. Once this phenomenon was discovered we could determine the rules of the toxicity by testing all 4096 possible 6mer seeds in a neutral backbone (after disabling passenger strand loading). We found that there is a good correlation between the toxicity of the shRNAs derived from the ORF of CD95L and the new 6mer seed toxicity (see Figure 1D in (Gao et al., 2018)). We therefore feel that the focus on CD95L in this research advance is well justified.

3) Similar to the above point, the authors could interrogate their existing data on the presence of additional small RNAs in the total RNA fraction used in Figure 4C to ask how many additional endogenous genes end up being processed. They could also interrogate the Ago pulldown data to see if they get incorporated into the Drosha KO RISC. Is there something special about CD95L mRNA processing, or does RISC load up with spurious RNA fragments in Drosha KOs? This is not clear from the analysis as presented. They already document the presence of VTRNA2-1, VTRA1-2 and TFCP2L1 sequences in the RISC – is there any reason to think these sequences might be toxic, or have additional cellular consequences?

This is an excellent suggestion made by the reviewer. Overexpression of CD95L mRNA caused loading of the RISC with small miRNA-sized RNAs derived from at least 22 regions in the CD95L ORF. This was most pronounced in Drosha k.o. cells devoid of most miRNAs. To determine whether this phenomenon could also be observed with endogenous coding mRNAs, we have now reanalyzed our small RNA Seq data of both total RNA and AGO bound RNA in HCT116 Drosha k.o. cells (infected with either empty vector or pLentiCD95L) as suggested by the reviewers. We identified all coding mRNAs that could give rise to sRNAs that were present in the AGO2 bound fraction with at least 10 reads. We only considered reads that uniquely aligned with their gene. While we found that the majority of protein coding mRNAs are not processed and loaded in the RISC in that fashion, about 3% of the coding genes in the genome are. Further analysis revealed that first, it is not only the expression level that determines whether a mRNA is processed (new Figure 5C). A number of very highly expressed mRNAs were not processed at all. Second, it demonstrated that CD95L was not just processed because it was a highly expressed exogenous gene. In fact, in the list of the most abundant RISC bound coding gene derived small RNAs, CD95L was number 14 from the top (see Figure 5D). Compared to the exogenously expressed CD95L the endogenously expressed mRNAs were processed in the same way. They were processed before being bound to the RISC but were further trimmed when bound to the RISC (new Figure 5A and Figure 5—figure supplement 1A). We are showing the top 10 genes with the highest read number bound to AGO proteins and the most read sites in their mRNAs represented in new Figure 5B and Figure 5—figure supplement 1B. Given the fact that not all mRNAs were processed in this way, we then asked whether the ones that were had specific cellular functions and the result was striking. We found a very strong enrichment for genes involved in cell proliferation (including cyclin D1 and MYC), and particularly for genes involved in protein translation. The majority of all mRNAs coding for ribosomal proteins (76%) were represented. While it is not clear how the sRNAs derived from these genes regulate cell function, the selective enrichment for genes of similar ontology to be RISC bound points at a biological function. We also find it intriguing that the gene ontology annotations that genes fall into are quite similar to the ones we found downregulated in cells undergoing DISE (Gao et al., 2018; Putzbach et al., 2018).

To obtain a preliminary sense of what sRNAs might be doing in the RISC we compared Drosha k.o. cells lacking most miRNAs with wt cells. We noticed that Drosha k.o. cells grow less well than wt cells (new Figure 5E). To test whether RISC bound sRNAs could negatively affect cell growth, we calculated the average seed toxicity of all sRNAs bound to Ago proteins and compared them to the average seed toxicity of the sRNAs bound to the RISC in wt cells (new Figure 5F). It was substantially different providing a possible explanation for the reduced growth rate of the k.o. cells. To directly test this we knocked down AGO2 (or AGO1-4, data not shown) to disable the RISC. This resulted in the cells growing as fast as the wt cells, suggesting that it is indeed RISC mediated RNAi that negatively affects cell growth. This experiment provides evidence of endogenous sRNAs being loaded into the RISC to negatively affect cell growth through the 6mer seed toxicity mechanism.

4) In Figure 1—figure supplement 2, I was surprised to see that the "SIL" mutant was as active as the wt CD95L mRNAs, despite expressing >85% less protein and having ~37% (somewhat evenly distributed) nucleotide divergence. Is this due to expression of specific sRNAs that were not altered by codon editing (maybe answerable by sRNA sequencing)? Is the ~12% of protein providing this activity (maybe answerable by removing the start codon)? Subjecting this mutant construct to the secondary structure prediction could be informative. The authors should reconcile this counterintuitive result.

Following the reviewers suggestion we have performed a secondary structure prediction of the SIL mutant. It was however, not very informative as most mRNAs when folded have stem loop structures (data not shown). However, now that we know that small RISC-bound RNAs derived from coding genes can kill cancer cells through 6mer seed toxicity, we compared the WT and the SIL mutant by plotting the 6mer seed toxicity along their sequences to compare their ability to give rise to toxic sRNAs. While there are many sequence differences in the RNA, the predicted 6mer seed toxicity in a number of areas are similar in both constructs (data not shown) and when displaying the toxicity of all sRNAs in a density plot (new Figure 1—figure supplement 3D) the difference between the two constructs did not reach statistical significance confirming that the SIL mutant could still give rise to multiple toxic sRNAs. Finally, in new Figure 3—figure supplement 1A (right panel) we demonstrate that similar to WT CD95L the SIL mutant mRNA's toxicity is dependent on AGO2, suggesting that the toxicity is exerted through RNAi.

5) The authors identify >20 potential segments within the CD95L mRNA that correspond to sRNAs sequenced after AGO IP. Similar to the experiments shown in Figure 1A, the authors should express (in HeyA8 cells) fragments of the CD95L mRNA (100-200 nts?, overlapping with one-another) in order to identify regions with the most toxic profile. If sensitization is necessary to see a phenotype, this experiment could be performed in the HCT116 wt and Drosha KO models. This is critical to understand if specific sequences (derived from the mRNA, not expressed separately via shRNA) direct the phenotype, and if local or broad RNA structure impacts sRNA biogenesis.

Before testing smaller regions derived from CD95L we generated and tested two CD95L fragments, a 5' and a 3' fragment to test whether folding of full length CD95L was required to produce toxic sRNAs. When expressed in HeyA8 CD95 k.o. cells the 5' fragment, which did not produce any detectable protein, was somewhat toxic but much less than full length CD95L (Figure 4—figure supplement 2A-C). The 3' fragment did not show toxicity.

In addition, we expressed the two fragments in HCT116 cells (Figure 4—figure supplement 2D and 2E).

When compared to the NP mutant neither of the two fragments caused any toxicity in wt HCT116 cells (Figure 4—figure supplement 2E, left panel). Both CD95L fragments, however, showed moderate toxicity in the hypersensitive Drosha k.o. cells (Figure 4—figure supplement 2E, right panel). In these cells, the full length NP mutant was quite toxic. We feel that the analysis of the exact folding rules of CD95L and its fragments is beyond the scope of this manuscript. However, these data suggest that while the local folding environment is important, folding of the entire wt mRNA seems to favor generation of toxic sRNAs.